# Dimensionality Reduction with Point-Distribution Similarity Invariant

**Hang Zhang** [1][2]  **Kai Ming Ting** [1][2]

## Abstract

Existing dimensionality reduction methods all perform dimensionality reduction by preserving some invariant in the space before and after dimensionality reduction. This paper proposes a new dimensionality reduction invariant: preserving the invariant of the point-distribution similarity. We also design a linear-time and effective method to achieve dimensionality reduction while preserving this invariant. We theoretically prove the feasibility of our method for dimensionality reduction. Furthermore, our results on benchmark datasets and single-cell expression data demonstrate the effectiveness and efficiency of the proposed method.

## 1. Introduction

Dimensionality reduction, as a crucial technique in machine learning, plays a vital role in many fields, serving as a fundamental method for data visualization and preprocessing (Van Der Maaten et al., 2009; Anowar et al., 2021). For example, it is used in atomistic simulations and general analysis of high-dimensional systems in physical chemistry (Rydzewski, 2023; Coifman et al., 2008), and as a preprocessing technique in single-cell domains to reduce the dimensionality of high-dimensional gene expression data for subsequent analysis (Shang & Zhou, 2022; Becht et al., 2019; Zhang et al., 2025b).

Existing and widely used dimensionality reduction algorithms, such as Principal Component Analysis (PCA) (Abdi & Williams, 2010), t-Distributed Stochastic Neighbor Embedding (t-SNE) (Maaten & Hinton, 2008), UMAP (McInnes et al., 2018), MultiDimensional Scaling (MDS) (Kruskal & Wish, 1978), and AutoEncoder (AE) (Wang et al., 2016), although very different, share a common char-

acteristic: *they preserve some invariant before and after dimensionality reduction.* For example, PCA preserves the dimensions with the maximum variance, t-SNE preserves the probability distribution of point pairs, UMAP preserves the topological structure, and AE preserves the reconstructed variables after dimensionality reduction. Except for PCA, which considers the variables unchanged and is therefore linear, the other algorithms consider the relationships between points unchanged and thus often have a quadratic time complexity (Van Der Maaten et al., 2009).

Existing algorithms often overlook the heterogeneous nature of real-world datasets, implicitly assuming a single generative source. In practice, however, data typically arises from a mixture of $k$ distinct distributions. For example, in classification and clustering datasets, the points in the same class (cluster) can be considered as being generated from the same distribution, $k$ classes (clusters) correspond to $k$ different distributions. Consequently, for downstream tasks such as classification, the preservation of pairwise point-to-point similarities is secondary to maintaining the affinity between a data point and these distributions. This observation motivates our central research question: *Is it possible to design a dimensionality reduction technique that explicitly preserves the similarity structure between data points and these distributions?*

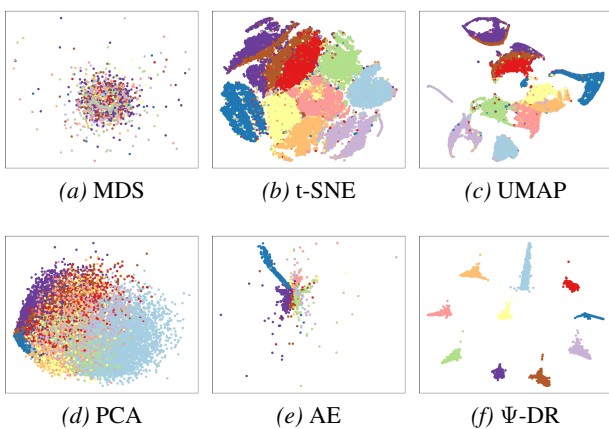

*Figure 1.* Comparison of dimensionality reduction results on the MNIST dataset. The first row has a complexity of quadratic or higher, while the second row has linear complexity.

[1]State Key Laboratory of Novel Software Technology, Nanjing University, Nanjing, China. [2]School of Artificial Intelligence, Nanjing University, Nanjing, China. Correspondence to: Kai Ming Ting <tingkm@nju.edu.cn>.

*Proceedings of the $43^{rd}$ International Conference on Machine Learning*, Seoul, South Korea. PMLR 306, 2026. Copyright 2026 by the author(s).

In the absence of label information, we approximate the $k$ underlying distributions by first employing a clustering algorithm to partition the data, followed by Kernel Mean Embedding (Muandet et al., 2017) for distribution estimation. Figure 1 presents a comparative analysis of our proposed dimensionality reduction method, $\Psi$-DR (or PSIDR), against baseline approaches on the MNIST dataset. By preserving point-distribution similarity invariants, $\Psi$-DR successfully disentangles the ten distinct digit distributions. Consequently, the resulting embeddings are significantly more conducive to downstream tasks such as classification and clustering.

The main contributions are summarized as follows:

1. Introducing a novel invariant for dimensionality reduction, defined by the *similarity between point and distribution*, rather than the traditional *point-to-point similarity*.

2. Proposing $\Psi$-DR, an effective linear-time framework that optimizes dimensionality reduction and maintains the integrity of this invariant.

3. Validating the effectiveness and efficiency of our method through theoretical analysis and extensive empirical experiments on benchmark datasets and single-cell expression data.

## 2. Related Work

Dimensionality reduction techniques aim to map high-dimensional data into a lower-dimensional manifold while preserving specific dimensionality reduction invariants. Based on the nature of the preserved invariant, we categorize existing literature into three main streams: variance-based, pairwise-similarity-based, and reconstruction-based invariant methods.

The most fundamental invariant in classical statistical analysis is the global variance, which represents the information content of the data. PCA (Pearson, 1901) is the pioneer of this category, preserving the maximum variance via orthogonal projection. Many other variants of PCA have been proposed to address the interpretation difficulty of PCA's dense loading vectors(Zou et al., 2006), to overcome the linearity limitation of PCA (Schölkopf et al., 1997), and to be robust to outliers (Candès et al., 2011).

A vast body of research focuses on preserving the geometric relationships between data points. These methods assume that the data lies on a low-dimensional manifold embedded in the high-dimensional space. MDS (Cox & Cox, 2008) preserves global Euclidean distances. Isomap (Tenenbaum et al., 2000) extends MDS by replacing Euclidean distances with geodesic distances to capture the true manifold geometry. On the other hand, Locally Linear Embedding (LLE) (Roweis & Saul, 2000) and Laplacian Eigenmaps (Belkin & Niyogi, 2003) focus on local invariants. t-SNE (Maaten & Hinton, 2008) minimizes the Kullback-Leibler divergence between joint probabilities in high and low dimensions, excelling at visualization but struggling with global structure. UMAP (McInnes et al., 2018) improves upon this by utilizing Riemannian geometry and fuzzy simplicial sets to better balance local and global structure preservation.

With the rise of deep learning, preserving the reconstruction capability, the ability to recover the original input from the latent space, has become a dominant paradigm. The standard Autoencoder (Hinton & Salakhutdinov, 2006) learns a compressed representation by minimizing the reconstruction error. Many variations were proposed afterwards (Vincent et al., 2008; Kingma & Welling, 2013; Moor et al., 2020).

Despite the diversity of these methods, they predominantly focus on either global statistics, pairwise point-to-point relations ($x_i \leftrightarrow x_j$), or self-reconstruction ($x_i \leftrightarrow \hat{x}_i$). This paper proposes a new invariant based on the similarity between point and $k$ distribution ($x_i \leftrightarrow \mathcal{P}_k$) in the data.

## 3. Problem Formulation

Let $\mathcal{X} \subset \mathbb{R}^d$ be a compact input space, and let $\mathcal{P}(\mathcal{X})$ denote the set of Borel probability measures on $\mathcal{X}$. Suppose the observed data $\{x_i\}_{i=1}^n \subset \mathcal{X}$ are generated from a mixture of $k$ underlying probability distributions $\{\nu_c\}_{c=1}^k \subset \mathcal{P}(\mathcal{X})$, i.e.,

$$x_i \sim \nu_{c(i)}, \quad c(i) \in \{1, \ldots, k\}.$$

Our goal is to learn a low-dimensional embedding $z_i \in \mathbb{R}^p$ ($p \ll d$) for each $x_i$ such that the **similarity structure among the underlying distributions** $\{\nu_c\}$ **and between points and distributions** is faithfully preserved.

Formally, let $\rho : \mathcal{P}(\mathcal{X}) \times \mathcal{P}(\mathcal{X}) \to [0,1]$ be a similarity function satisfying $\rho(\nu, \nu') = \langle \nu, \nu' \rangle_\mathcal{H}$ in the Hilbert space $\mathcal{H}$, and let $\sigma : \mathcal{X} \times \mathcal{P}(\mathcal{X}) \to [0,1]$ denote the point–distribution similarity, defined as $\sigma(x, \nu) = \mathbb{E}_{x' \sim \nu}[\kappa(x, x')]$ for some symmetric positive definite kernel $\kappa$. We seek an embedding map $f : \mathcal{X} \to \mathbb{R}^p$ and a set of centers $\{\bar{z}_c\}_{c=1}^k \subset \mathbb{R}^p$ such that for all $i$ and $c$,

$$\langle f(x_i), \bar{z}_c \rangle_{\mathbb{R}^p} \approx \sigma(x_i, \nu_c),$$
$$\text{and} \quad \|\bar{z}_c - \bar{z}_{c'}\|_2^2 \approx \|\nu_c - \nu_{c'}\|_\mathcal{H}^2.$$

This leads to the following **point-distribution similarity invariant-preserving dimensionality reduction** problem:

*Problem* 3.1. Given the dataset $\{x_i\}_{i=1}^n$ and the set of underlying distributions $\{\nu_c\}_{c=1}^k$, our objective is to find an embedding map $f : \mathcal{X} \to \mathbb{R}^p$ and a set of centers

$\{\bar{z}_c\}_{c=1}^k \subset \mathbb{R}^p$ (where $p \ll d$) that minimize the following objective function:

$$\mathcal{L}(f, \{\bar{z}_c\}) = \sum_{i=1}^n \sum_{c=1}^k \left(\langle f(x_i), \bar{z}_c \rangle - \sigma(x_i, \nu_c)\right)^2$$
$$+ \lambda \sum_{c,c'=1}^k \left(\|\bar{z}_c - \bar{z}_{c'}\|_2^2 - \|\nu_c - \nu_{c'}\|_{\mathcal{H}}^2\right)^2,$$

subject to $p \ll d$, where $\lambda > 0$ balances point–distribution and inter-distribution similarity preservation.

However, Problem 3.1 is intractable in practice: the true distributions $\{\nu_c\}$ are unknown, and the similarity functionals $\sigma$ and $\|\cdot\|_{\mathcal{H}}$ cannot be evaluated without access to the full distributions.

To render the problem computable, we represent each distribution $\nu_c$ via its **kernel mean embedding** (KME) in a reproducing kernel Hilbert space (RKHS) (Berlinet & Thomas-Agnan, 2011).

Let $k : \mathcal{X} \times \mathcal{X} \to \mathbb{R}$ be a continuous, bounded, positive definite kernel with RKHS $\mathcal{H}_\kappa$ and feature map $\phi : \mathcal{X} \to \mathcal{H}_\kappa$, such that $\kappa(x, x') = \langle \phi(x), \phi(x') \rangle_{\mathcal{H}_\kappa}$. The kernel mean embedding of $\nu_c$ is defined as: $\mu_c := \mathbb{E}_{x \sim \nu_c}[\phi(x)] \in \mathcal{H}_\kappa$.

Under mild conditions (e.g., $\kappa$ being characteristic), the map $\nu \mapsto \mu_\nu$ is injective, ensuring that the distributional information is preserved in $\mathcal{H}_\kappa$.

Given empirical samples $\{x_i\}_{i \in C_c}$ from subset $C_c$ that drawn from $\nu_c$, we approximate $\mu_c$ by the empirical mean embedding: $\hat{\mu}_c := \frac{1}{|C_c|} \sum_{i \in C_c} \phi(x_i)$.

Crucially, the point–distribution similarity becomes

$$\sigma(x_i, \nu_c) \approx \langle \phi(x_i), \hat{\mu}_c \rangle_{\mathcal{H}_\kappa} = \frac{1}{|C_c|} \sum_{j \in C_c} k(x_i, x_j),$$

and the squared distance between distributions is

$$\|\nu_c - \nu_{c'}\|_{\mathcal{H}_\kappa}^2 \approx \|\hat{\mu}_c - \hat{\mu}_{c'}\|_{\mathcal{H}_\kappa}^2$$
$$= \langle \hat{\mu}_c, \hat{\mu}_c \rangle + \langle \hat{\mu}_{c'}, \hat{\mu}_{c'} \rangle - 2\langle \hat{\mu}_c, \hat{\mu}_{c'} \rangle.$$

Substituting these empirical estimates into Problem 3.1, we obtain the following **kernel mean embedding-based similarity-preserving DR** problem:

*Problem* 3.2. Let $\{C_c\}_{c=1}^k$ be a subset of $\{1, \ldots, n\}$. Find $Z = [z_1, \ldots, z_n]^\top \in \mathbb{R}^{n \times p}$ and $\{\bar{z}_c\}_{c=1}^k \subset \mathbb{R}^p$ that minimize

$$\mathcal{L}_{\text{KME}}(Z, \{\bar{z}_c\}) = \sum_{i=1}^n \sum_{c=1}^k \left(\langle z_i, \bar{z}_c \rangle - \langle \phi(x_i), \hat{\mu}_c \rangle_{\mathcal{H}_\kappa}\right)^2$$
$$+ \lambda \sum_{c,c'=1}^k \left(\|\bar{z}_c - \bar{z}_{c'}\|_2^2 - \|\hat{\mu}_c - \hat{\mu}_{c'}\|_{\mathcal{H}_\kappa}^2\right)^2.$$

Problem 3.2 is now fully data-driven and computable via a kernel function. The proposed algorithm solves Problem 3.2 through a decoupled two-stage procedure:

(i) embedding [1] distributions via MDS on $\|\hat{\mu}_c - \hat{\mu}_{c'}\|_{\mathcal{H}_\kappa}$.

(ii) locating the position of each point via multilateration (Widdison & Long, 2024) using distances derived from $\langle \phi(x_i), \hat{\mu}_c \rangle_{\mathcal{H}_\kappa}$.

# 4. Optimization Problem Decomposition and Solution Strategy

The joint optimization Problem 3.2 is non-convex and high-dimensional. We adopt a **decoupled two-stage strategy** that first embeds the distributions $\hat{\mu}_c$ and then locating the position of each data point with respect to $\hat{\mu}_c$. This approach is not only computationally efficient but also theoretically well-motivated, as it directly controls the primary objective of preserving the distance between two distributions: $\|\hat{\mu}_c - \hat{\mu}_{c'}\|_{\mathcal{H}_\kappa}$ before addressing the point-level fidelity: $\langle \phi(x_i), \hat{\mu}_c \rangle_{\mathcal{H}_\kappa}$.

Since Problem 3.2 requires dividing the data into $k$ subsets, to ensure the effectiveness and efficiency of the division, we use the recently proposed linear-time clustering method KBC (Zhang et al., 2025a) to obtain $k$ subsets. Then, our method decomposes Problem 3.2 into the following sequential subproblems:

We first isolate the inter-distribution term in $\mathcal{L}_{\text{KME}}$. Since the empirical kernel mean embeddings $\{\hat{\mu}_c\}_{c=1}^k$ are now fixed, the problem of finding their low-dimensional counterparts $\{\bar{z}_c\}_{c=1}^k$ reduces to a **classical multidimensional scaling (MDS)** problem.

From the kernel mean embeddings, we compute the empirical squared MMD (Maximum Mean Discrepancy) matrix $\mathbf{D} \in \mathbb{R}^{k \times k}$, where $D_{cc'} = \|\hat{\mu}_c - \hat{\mu}_{c'}\|_{\mathcal{H}_\kappa}^2$. Classical MDS seeks a configuration of points $\{\bar{z}_c\}_{c=1}^k \subset \mathbb{R}^p$ whose Euclidean distances best match $\mathbf{D}$. This is achieved by:

$$\min_{\{\bar{z}_c\} \subset \mathbb{R}^p} \sum_{c,c'=1}^k \left(\|\bar{z}_c - \bar{z}_{c'}\|_2^2 - D_{cc'}\right)^2. \tag{1}$$

This problem admits a closed-form solution via an eigendecomposition of the double-centered distance matrix, ensuring that the global structure of the distributional space is preserved in $\mathbb{R}^p$.

**Locating Data Points via Multilateration**

With the low-dimensional centers $\{\bar{z}_c\}_{c=1}^k$ fixed from Stage 1, we now solve for the individual point embed-

---

[1]To avoid ambiguity, with the exception of 'kernel mean embedding', the term 'embedding' refers to mapping points from a high-dimensional space to a low-dimensional space.

dings $\{z_i\}_{i=1}^n$. We first convert the point-distribution inner products from Problem 3.2 into distances, which is more amenable to Multilateration.

The point-distribution similarity $\langle \phi(x_i), \hat{\mu}_c \rangle_{\mathcal{H}_\kappa}$ can be converted to an approximate squared distance in the embedding space using: $d_{ic}^2 \approx 2(1 - \langle \phi(x_i), \hat{\mu}_c \rangle_{\mathcal{H}_\kappa})$, assuming that the embeddings are normalized[2]. The problem of locating the position of each point $z_i$ can be achieved by computing its distances to the fixed anchors $\{\bar{z}_c\}$ that match the target distances $\{d_{ic}\}$. This is the well-known **multilateration problem** (Widdison & Long, 2024; Norrdine, 2012):

$$z_i = \arg\min_{z \in \mathbb{R}^p} \sum_{c=1}^k \left( \|z - \bar{z}_c\|_2 - d_{ic} \right)^2. \quad (2)$$

For $k > p + 1$, this is an over-determined system. We use the closed-form least-squares solution derived from squaring the distance equations and canceling quadratic terms. This yields a linear system $Az = b_i$, which can be solved efficiently via the normal equations. In degenerate cases (e.g., anchors are co-planar in a way that makes $A$ rank-deficient), we fall back to a robust numerical optimizer[3].

**Post-processing: Subset Center Alignment**

The two-stage procedure does not guarantee that the empirical mean of the embedded points in subset $c$ will coincide with its target center $\bar{z}_c$. To enforce this consistency post-hoc, we perform a final rigid translation for each subset: $z_i \leftarrow z_i + (\bar{z}_c - \frac{1}{|C_c|} \sum_{j \in C_c} z_j), \quad \forall i \in C_c$.

This step is optional in practice, as it is not a crucial part of maintaining dimensionality reduction invariant (point-distribution similarity).

This pipeline provides a highly efficient and effective approximate solution to the intractable joint Problem 3.2, with each stage being either closed-form or involving a simple, well-conditioned optimization.

The proposed $\Psi$-DR is shown in Algorithm 1 [4].

**Time Complexity Analysis:** Let $n$ be the number of data points, $k$ be the number of distributions, and $p$ be the target dimension ($p \ll d$).Denote $\check{c} = \max_c |C_c|$ as the size of the largest subset.

We analyze the time complexity of the proposed $\Psi$-DR algorithm across its four stages. The preprocessing phase, comprising KBC clustering and the computation of $k$ ker-

---

**Algorithm 1** $\Psi$-DR (or PSIDR): **P**oint-distribution **S**imilarity **I**nvariant **D**imensionality **R**eduction

**Require:** Data $\mathcal{X} \in \mathbb{R}^{n \times d}$, number of distributions $k$, target number of dimensions $p$, kernel $k(\cdot, \cdot)$.
**Ensure:** Embedding $\mathbf{Z} \in \mathbb{R}^{n \times p}$
    **Obtain subsets $C_c$ and kernel mean embedding**
1: $\{C_c\}_{c=1}^k \leftarrow \text{KBC}(\mathcal{X}; k)$
2: $\hat{\mu}_c = \frac{1}{|C_c|} \sum_{i \in C_c} \phi(x_i), \quad \forall c = 1, \dots, k$
    **Embedding distributions**
3: $\mathbf{D} \leftarrow D_{cc'} = \|\hat{\mu}_c - \hat{\mu}_{c'}\|_{\mathcal{H}_\kappa}^2$
4: $\{\bar{z}_c\}_{c=1}^k \leftarrow \text{MDS}(\mathbf{D}, p)$
    **Locating each point via Multilateration**
5: **for** $i = 1$ **to** $n$ **do**
6:     $d_{ic} = \sqrt{2(1 - \langle \phi(x_i), \hat{\mu}_c \rangle)}$
7:     $z_i \leftarrow \arg\min_{z \in \mathbb{R}^p} \sum_{c=1}^k \left( \|z - \bar{z}_c\|_2 - d_{ic} \right)^2$
8: **end for**
    **Post-processing (optional)**
9: **for** $c = 1$ **to** $k$ **do**
10:     $\hat{\mu}_c \leftarrow \frac{1}{|C_c|} \sum_{j \in C_c} z_j$
11:     $z_i \leftarrow z_i + (\bar{z}_c - \hat{\mu}_c), \quad \forall i \in C_c$
12: **end for**
13: **Return** $\mathbf{Z} = \{z_i\}_{i=1}^n$

---

nel mean embeddings, incurs a cost of $O(nkd) + O(\bar{c})$. Subsequently, constructing the pairwise MMD matrix $\mathbf{D}$ requires $O(k^2)$ operations. The embedding of distributions relies on the classical MDS, which involves an eigendecomposition with a complexity of $O(k^3)$. In the multilateration phase, computing target distances takes $O(nk)$, while solving the closed-form multilateration problem for $n$ points scales as $O(np^2 k)$. Finally, the post-processing adds a negligible cost of $O(np)$. Consequently, the overall time complexity is $O(k^3 + np^2 k + nkd)$. Since the number of distributions $k$ and the target dimension $p$ are typically much smaller than the sample size $n$ ($k, p \ll n$), the algorithm achieves linear scalability with respect to $n$ and $d$.

## 5. Theoretical Analysis

We now provide a statistical analysis of Algorithm 1. Our main result (Theorem 5.7) establishes that the learned embedding is *consistent* in the sense that it converges in mean squared error to an ideal similarity-preserving configuration as the sample size grows. The proof proceeds through three key lemmas that carefully quantify the estimation error, metric distortion, and sensitivity in multilateration. The lemmas and theorems are summarized in Table 1.

We begin by stating and motivating our assumptions.

**Assumption 5.1** (Bounded and Characteristic Kernel). The kernel $\kappa : \mathcal{X} \times \mathcal{X} \to \mathbb{R}$ is continuous, positive definite, and bounded: there exists $\varepsilon > 0$ such that

---

[2]Throughout this paper, $d_{ic}$ denotes the scalar target distance between the $i$-th data point and the $c$-th distribution. We define $\mathbf{d}_i = [d_{i1}, \dots, d_{ik}]^\top \in \mathbb{R}^k$ as the vector collecting these distances for point $i$. In the theoretical analysis (e.g., Lemma 5.6), we drop the index $i$ and use $\mathbf{d}$ to denote a generic distance vector.

[3]See the details in Appendix.

[4]The source code is available in https://github.com/IsolationKernel/PSIDR.

*Table 1.* Summary of the theoretical results for the point-distribution similarity invariant dimensionality reduction.

| Result | Description | Reference |
|---|---|---|
| KME Concentration | Empirical KME converge uniformly to true means at rate $O(1/\sqrt{n_{\min}})$. | Lem. 5.4 |
| MDS Stability | Center of embedding is stable under distance matrix perturbations. | Lem. 5.5 |
| Multilateration Sensitivity | The multilateration is Lipschitz continuous in distances and anchors. | Lem. 5.6 |
| Embedding Consistency | Learned embedding converges to ideal configuration as $n \to \infty$. | Thm. 5.7 |
| Finite-Sample Error Bound | Non-asymptotic bound on average embedding error. | Thm. 5.8 |
| Distance Preservation | Point-to-distribution distances in $\mathbb{R}^p$ consistently estimate RKHS distances. | Thm. 5.9 |

$\sup_{x \in \mathcal{X}} \kappa(x, x) \leq \varepsilon$. Moreover, $\kappa$ is characteristic, so the mean embedding map $\nu \mapsto \mu_\nu = \mathbb{E}_{x \sim \nu}[\phi(x)]$ is injective.

This assumption is standard in kernel methods (Berlinet & Thomas-Agnan, 2011) and ensures that the feature map $\phi(x)$ has uniformly bounded norm, which is crucial for the concentration inequalities. The characteristic property guarantees that distinct distributions have distinct embeddings, which is fundamental to preserving distributional information in RKHS $\mathcal{H}_\kappa$. Without this, different distributions could be mapped to the same point in $\mathcal{H}_\kappa$, making similarity preservation impossible.

**Assumption 5.2** (Asymptotically Balanced Subsets). There exists a partition $\{C_c\}_{c=1}^k$ of $\{x_1, \dots, x_n\}$ such that $n_{\min} := \min_{c \in [k]} |C_c| \to \infty$ as $n \to \infty$.

This assumption ensures that every subset receives sufficient data to accurately estimate its kernel mean embedding. It rules out pathological cases where a subset contains only a constant number of points, which would lead to inconsistent KME estimates. In practice, clustering algorithms like KBC (used in our method) tend to produce reasonably balanced subsets under mild data regularity conditions, making this assumption realistic for large datasets.

**Assumption 5.3** (Non-degenerate Limit Geometry). Let $\mathbf{D}^* \in \mathbb{R}^{k \times k}$ with $D_{cc'}^* = \|\mu_c - \mu_{c'}\|_{\mathcal{H}_\kappa}^2$, and let $\mathbf{B}^* = -\frac{1}{2}\mathbf{H}\mathbf{D}^*\mathbf{H}$, where $\mathbf{H} = \mathbf{I}_k - \frac{1}{k}\mathbf{1}_k\mathbf{1}_k^\top$. Denote the eigenvalues of $\mathbf{B}^*$ by $\lambda_1^* \geq \dots \geq \lambda_k^* \geq 0$. Assume that for the target dimension $p < k$, the eigengap $\Delta_p := \lambda_p^* - \lambda_{p+1}^*$ satisfies $\Delta_p > 0$. Furthermore, assume the matrix of true MDS centers $\bar{\mathbf{Z}}^* \in \mathbb{R}^{k \times p}$ has full column rank.

This geometric assumption ensures that the true low-dimensional configuration of centers is identifiable and stable. A positive eigengap $\Delta_p$ guarantees that the top-$p$ eigenspace of the Gram matrix is well-separated from the rest, which is a standard requirement for the consistency of spectral methods like MDS (see, e.g., Von Luxburg et al. (2008)). The full-rank condition on $\bar{\mathbf{Z}}^*$ prevents degenerate configurations (e.g., all centers lying on a lower-dimensional subspace), which would make the multilateration problem ill-posed.

With these assumptions in place, we now establish the key

technical lemmas.

**Lemma 5.4** (Uniform Concentration of Kernel Mean Embeddings). *Suppose Assumption 5.1 holds. For any $\delta \in (0, 1)$, with probability at least $1 - \delta$,*

$$\max_{c \in [k]} \|\hat{\mu}_c - \mu_c\|_{\mathcal{H}_\kappa} \leq \sqrt{\frac{\varepsilon}{n_{\min}}}\left(1 + \sqrt{6 \log \frac{2k}{\delta}}\right).$$

Lemma 5.4 quantifies the uniform estimation error of the empirical kernel mean embeddings across all $k$ distributions. The $O(1/\sqrt{n_{\min}})$ rate is minimax optimal and matches the parametric rate for mean estimation in Hilbert spaces. The logarithmic dependence on $k$ arises from the union bound and is unavoidable when controlling the error uniformly over multiple distributions.

**Lemma 5.5** (Perturbation Bound for MDS Embeddings). *Let $\mathbf{D}, \mathbf{D}^* \in \mathbb{R}^{k \times k}$ be Euclidean distance matrices, and let $\bar{\mathbf{Z}}, \bar{\mathbf{Z}}^* \in \mathbb{R}^{k \times p}$ be their classical MDS embeddings (i.e., $\bar{\mathbf{Z}}\bar{\mathbf{Z}}^\top = \mathbf{B}$, where $\mathbf{B} = -\frac{1}{2}\mathbf{H}\mathbf{D}\mathbf{H}$, and similarly for $\bar{\mathbf{Z}}^*$). Let $\mathcal{O}(p)$ denote the set of $p \times p$ orthogonal matrices. Under Assumption 5.3,*

$$\min_{\mathbf{Q} \in \mathcal{O}(p)} \|\bar{\mathbf{Z}} - \bar{\mathbf{Z}}^*\mathbf{Q}\|_F \leq \frac{\sqrt{2p}}{2\Delta_p}\|\mathbf{D} - \mathbf{D}^*\|_F.$$

Lemma 5.5 formalizes the stability of the MDS step: small perturbations in the input distance matrix lead to small perturbations in the output embedding, up to an orthogonal transformation. The bound depends inversely on the eigengap $\Delta_p$, highlighting that well-separated eigenvalues (a non-degenerate spectrum) are crucial for stability. This result bridges the statistical error from Lemma 5.4 (which affects $\mathbf{D}$) to geometric error in the centers $\bar{\mathbf{Z}}$.

**Lemma 5.6** (Multilateration Sensitivity). *Let $\bar{\mathbf{Z}} = [\bar{z}_1, \dots, \bar{z}_k]^\top \in \mathbb{R}^{k \times p}$ have full column rank, and define $\mathbf{A} = -2(\bar{\mathbf{Z}}_{2:k} - \mathbf{1}_{k-1}\bar{z}_1^\top) \in \mathbb{R}^{(k-1) \times p}$. Assume $\sigma_{\min}(\mathbf{A}) \geq \sigma_0 > 0$. Let $\mathbf{d} \in \mathbb{R}^k$ be distances, and define $\mathbf{b} \in \mathbb{R}^{k-1}$ by $b_c = d_{c+1}^2 - d_1^2 - (\|\bar{z}_{c+1}\|_2^2 - \|\bar{z}_1\|_2^2)$. Let $z^\dagger = \mathbf{A}^\dagger\mathbf{b}$[5]. Then for any two inputs $(\mathbf{d}, \bar{\mathbf{Z}})$ and $(\mathbf{d}', \bar{\mathbf{Z}}')$,*

$$\|z^\dagger - z^{\dagger'}\|_2 \leq L\left(\|\mathbf{d}^2 - \mathbf{d'}^2\|_2 + \|\bar{\mathbf{Z}} - \bar{\mathbf{Z}}'\|_F\right),$$

---

[5] $\dagger$ denotes the Moore-Penrose pseudoinverse (Barata & Hussein, 2012).

*where the Lipschitz constant $L = O\left(\frac{1}{\sigma_0^2}\left(1 + \frac{|\mathbf{b}|_2}{\sigma_0}\right)\right)$.*

This lemma captures the robustness of the closed-form multilateration procedure. The constant $L$ depends on the smallest singular value $\sigma_0$ of the anchor matrix $\mathbf{A}$, which measures how well-conditioned the multilateration problem is. When anchors are in general position (as ensured by Assumption 5.3), $\sigma_0$ is bounded away from zero, making the multilateration stable. This result ensures that errors in the target distances (from KME estimation) and anchor positions propagate linearly to the final point embeddings.

**Theorem 5.7** (Consistency of the Embedding). *Let $z_i^* \in \mathbb{R}^p$ satisfy $\|z_i^* - \bar{z}_c^*\|_2^2 = 2(1 - \langle \phi(x_i), \mu_c \rangle_{\mathcal{H}_\kappa})$ for all $c$. Let $z_i$ be the embedding, Then, as $n \to \infty$,*

$$\frac{1}{n}\sum_{i=1}^{n}\|z_i - z_i^*\|_2^2 \xrightarrow{\mathbb{P}} 0.$$

Theorem 5.7 states that the embedding produced by our algorithm converges to an ideal configuration that perfectly preserves the distributional similarities defined by the kernel mean embeddings. The proof combines the three lemmas in a clean error propagation argument: KME estimation error (Lemma 5.4) induces error in the MDS step (Lemma 5.5), which in turn affects multilateration (Lemma 5.6). The rate of convergence is $O(k^2/n_{\min})$, which is favorable when the number of distributions $k$ is fixed and subsets are balanced ($n_{\min} = \Theta(n/k)$).

**Theorem 5.8** (Finite-Sample Embedding Error Bound). *Under Assumptions 5.1, 5.2, and 5.3, there exists a constant $C > 0$, such that for any $\delta \in (0, 1)$, with probability at least $1 - \delta$,*

$$\frac{1}{n}\sum_{i=1}^{n}\|z_i - z_i^*\|_2^2 \leq C \cdot \frac{k^2}{n_{\min}}\left(1 + \log\frac{2k}{\delta}\right).$$

While Theorem 5.7 establishes asymptotic consistency, Theorem 5.8 provides a more practical guide by characterizing the convergence rate. Two key insights emerge from this bound: (i) The error bound scales inversely with $n_{\min}$, not the total sample size $n$. This highlights that the performance of $\Psi$-DR is determined by the least represented distribution. To achieve a low embedding error, the dataset should ideally be balanced; for highly imbalanced data, $n_{\min}$ becomes the bottleneck. (ii) The error scales quadratically with the number of distributions $k$. This reflects the cumulative uncertainty: as $k$ increases, the MDS stage (embedding $k$ distributions) accumulates more estimation errors from the $k \times k$ MMD matrix, which subsequently propagate to the multilateration stage.

**Theorem 5.9** (Point-to-Distribution Distance Preservation). *Assume the kernel is normalized ($k(x, x) = 1$ for*

all $x$). *Let $d_{ic}^{\mathcal{H}} = \|\phi(x_i) - \mu_c\|_{\mathcal{H}_\kappa}$ and $d_{ic}^{\mathbb{R}^p} = \|z_i - \bar{z}_c\|_2$. Under Assumptions 5.1, 5.2, and 5.3, for any $\delta \in (0, 1)$, exist $C' > 0$, with probability at least $1 - \delta$,*

$$\max_{i \in [n],\, c \in [k]}\left|d_{ic}^{\mathbb{R}^p} - d_{ic}^{\mathcal{H}}\right| \leq C' \cdot \frac{k}{\sqrt{n_{\min}}}\left(1 + \sqrt{\log\frac{2k}{\delta}}\right).$$

This result is of central importance as it directly validates the core motivation of our method. Recall that our objective (Problem 3.2) is to preserve the similarity between points and distributions. Theorem 5.9 guarantees that, with high probability, the Euclidean distances $\|z_i - \bar{z}_c\|_2$ in the low-dimensional space $\mathbb{R}^p$ are consistent estimates of the distances $\|\phi(x_i) - \mu_c\|_{\mathcal{H}_\kappa}$ in RKHS. This metric preservation property implies that downstream tasks, such as $k$-means, perform comparably in the linear embedding space as they would using the computationally expensive kernel methods in the original high-dimensional space. The convergence rate of $O(1/\sqrt{n_{\min}})$ is consistent with standard results for kernel mean estimation, indicating that our two-stage reduction procedure does not degrade the statistical efficiency of the underlying estimator.

# 6. Experiments

To demonstrate the effectiveness of our proposed $\Psi$-DR algorithm, we compared it with existing algorithms, PCA, t-SNE, UMAP, AE, and MDS on benchmark datasets and single-cell gene expression datasets. These datasets range in size from 1372 to 804414 samples and have dimensions from 3 to 47236. We set the target dimension to 2. The detailed information is provided in the appendix.

To evaluate intrinsic clustering structure and compactness without reliance on downstream tasks, we use the Davies–Bouldin Index (DB) (Davies & Bouldin, 2009) and the Calinski–Harabasz Index (CH) (Caliński & Harabasz, 1974). Lower DB scores and higher CH scores indicate better separation and cohesion of clusters, reflecting more effective dimensionality reduction. Since reduced representations are often used in downstream applications, we further assess their utility through supervised and unsupervised tasks. We perform $k$-means clustering on the embeddings and report Normalized Mutual Information (NMI) (Vinh et al., 2010) between predicted clusters and ground-truth labels. Additionally, we evaluate classification performance by training a random forest classifier (Breiman, 2001) on an 80% training split of the reduced data and reporting accuracy (ACC) on the remaining 20% test split. The experiments are conducted with the post-processing.

## 6.1. Comparison between different invariants

The result of the dimensionality reduction comparison is shown in Table 2. $\Psi$-DR outperforms other methods in

*Table 2.* Dimensionality Reduction Performance Comparison. The winner is shown in bold, the runner-up underlined.

| Dataset | Calinski-Harabasz (↑) | | | | | | Davies-Bouldin (↓) | | | | | |
|---|---|---|---|---|---|---|---|---|---|---|---|---|
| | PCA | t-SNE | UMAP | AE | MDS | Ψ-DR | PCA | t-SNE | UMAP | AE | MDS | Ψ-DR |
| COIL20 | 298 | 1284 | **4058** | 87 | 90 | 3985 | 6.84 | 3.82 | 4.01 | 4.99 | 8.54 | **0.72** |
| Banknote | 298 | 792 | 1369 | 562 | 342 | **37146** | 1.92 | 1.15 | 0.84 | 1.26 | 1.76 | **0.05** |
| Landsat | 1158 | 1244 | 1609 | 1118 | 826 | **1990** | 1.27 | 1.08 | 1.18 | 1.19 | 1.28 | **1.06** |
| Pendigits | 2270 | 4078 | 6528 | 1306 | 2092 | **7470** | 5.74 | 1.97 | 1.53 | 4.78 | 4.15 | **1.27** |
| USPS | 1063 | 5286 | **9756** | 726 | 652 | 8252 | 5.62 | 1.77 | 1.42 | 18.93 | 8.75 | **1.05** |
| Letters | 339 | **660** | 193 | 388 | 344 | 418 | 20.34 | 13.73 | 17.66 | 19.31 | 27.34 | **11.92** |
| MNIST | 4986 | 30448 | 46927 | 3246 | 2 | **52306** | 8.47 | 1.48 | 1.46 | 5.51 | 191.87 | **0.96** |
| Skin | 33589 | 59854 | 39187 | 2632 | NA | **1279417** | 1.65 | 1.33 | 1.60 | 2.03 | NA | **0.13** |
| Gisette | 689 | 4911 | 7346 | 865 | 1 | **21224** | 2.70 | 1.07 | 0.89 | 1.83 | 102.46 | **0.34** |
| Rcv1 | 7199 | 5281 | NA | NA | NA | **15722** | 73.54 | 27.69 | NA | NA | NA | **15.73** |

| Dataset | Accuracy (↑) | | | | | | Normalized Mutual Information (↑) | | | | | |
|---|---|---|---|---|---|---|---|---|---|---|---|---|
| | PCA | t-SNE | UMAP | AE | MDS | Ψ-DR | PCA | t-SNE | UMAP | AE | MDS | Ψ-DR |
| COIL20 | 0.729 | 0.941 | 0.865 | 0.896 | 0.795 | **0.997** | 0.626 | 0.864 | 0.839 | 0.622 | 0.626 | **0.960** |
| Banknote | 0.876 | **1.000** | 0.996 | 0.989 | 0.935 | 0.996 | 0.011 | 0.349 | 0.793 | 0.646 | 0.011 | **0.936** |
| Landsat | 0.835 | **0.900** | 0.865 | 0.868 | 0.843 | 0.855 | 0.537 | 0.634 | 0.618 | 0.619 | 0.542 | **0.681** |
| Pendigits | 0.662 | **0.996** | 0.992 | 0.968 | 0.806 | 0.906 | 0.545 | 0.811 | 0.795 | 0.600 | 0.581 | **0.819** |
| USPS | 0.379 | **0.952** | 0.910 | 0.873 | 0.372 | 0.825 | 0.309 | 0.673 | 0.741 | 0.414 | 0.256 | **0.799** |
| Letters | 0.242 | **0.944** | 0.887 | 0.654 | 0.258 | 0.607 | 0.186 | 0.488 | 0.437 | 0.259 | 0.201 | **0.513** |
| MNIST | 0.321 | 0.938 | 0.911 | 0.743 | 0.198 | **0.970** | 0.209 | 0.671 | 0.717 | 0.304 | 0.075 | **0.814** |
| Skin | 0.995 | **0.999** | **0.999** | 0.997 | NA | 0.996 | 0.014 | 0.006 | 0.202 | 0.005 | NA | **0.828** |
| Gisette | 0.644 | **0.961** | 0.954 | 0.937 | 0.525 | 0.931 | 0.070 | 0.630 | **0.702** | 0.149 | 0.001 | 0.655 |
| Rcv1 | 0.384 | **0.624** | NA | NA | NA | 0.547 | 0.148 | 0.128 | NA | NA | NA | **0.343** |

*Table 3.* Summary of the Comparison.

| Method | PCA | t-SNE | UMAP | AE | MDS | Ψ-DR |
|---|---|---|---|---|---|---|
| winner | 0 | 9 | 4 | 0 | 0 | 28 |
| runner-up | 1 | 12 | 19 | 1 | 0 | 6 |

*Table 4.* Compute time (in CPU seconds). For times exceeding 100 seconds, we report integer values.

| Dataset | PCA | t-SNE | UMAP | AE | MDS | Ψ-DR |
|---|---|---|---|---|---|---|
| COIL20 | **0.06** | 6.52 | 15.65 | 4.62 | 40.06 | 0.66 |
| Banknote | **0.01** | 6.53 | 3.01 | 3.90 | 14.12 | 0.15 |
| Landsat | **0.01** | 10.82 | 5.66 | 5.30 | 72.47 | 0.16 |
| Pendigits | **0.01** | 84.54 | 26.93 | 28.71 | 3437 | 2.19 |
| USPS | **0.03** | 81.90 | 15.13 | 28.24 | 3418 | 6.32 |
| Letters | **0.01** | 180 | 15.51 | 52.11 | 12110 | 10.47 |
| MNIST | **1.05** | 1030 | 117 | 184 | 19472 | 28.55 |
| Skin | **0.06** | 2207 | 1518 | 612 | NA | 22.71 |
| Gisette | **0.47** | 10.36 | 61.29 | 48.42 | 324 | 2.92 |
| Rcv1 | 549 | 35638 | NA | NA | NA | **382** |

terms of CH, DB, and NMI. t-SNE achieved the best results in terms of ACC.

We summarize the number of winners and runner-ups for each algorithm in Table 3, with Ψ-DR achieving the most winner counts. This demonstrates the effectiveness of our proposed method of using the point-distribution similarity as an invariant for dimensionality reduction, and the effectiveness of the Ψ-DR algorithm. This effectiveness is reflected not only in the dimensionality reduction performance using CH and DB as the evaluation metrics, but also in the effectiveness of downstream tasks such as unsupervised $k$-means clustering using NMI.

Table 4 shows the running time of different algorithms. PCA, as a well-known linear-time dimensionality reduction method, is very efficient in dimensionality reduction. Our proposed Ψ-DR, as a linear-time dimensionality reduction algorithm, is also very efficient, second only to PCA. It is worth noting that on the Rcv1 dataset, which has a very large number of points (804,414) and a very high original dimension (47,236), Ψ-DR is more efficient because it is linear with respect to the number of points and the original number of dimensions $d$, while PCA has the quadratic complexity with respect to $d$.

*Table 5.* Clustering Performance Comparison on single-cell expression data. The winner is shown in bold, the runner-up underlined.

| Dataset | Calinski-Harabasz (↑) | | | | | | Davies-Bouldin (↓) | | | | | |
|---|---|---|---|---|---|---|---|---|---|---|---|---|
| | PCA | t-SNE | UMAP | AE | MDS | Ψ-DR | PCA | t-SNE | UMAP | AE | MDS | Ψ-DR |
| airway | **9410** | 607 | 2564 | 1502 | 5 | 3727 | 2.62 | 5.92 | 3.05 | 1.29 | 25.43 | **2.13** |
| crohn | **13862** | 726 | 515 | 1196 | 2 | 8407 | 6.52 | 15.15 | 20.61 | 5.58 | 180.94 | **6.50** |
| tonsil | **1895** | 75 | 170 | 757 | 2 | 719 | 7.15 | 19.97 | 16.20 | **3.35** | 108.75 | 15.62 |
| jurkat | 1661 | 72 | 16 | 405 | 9 | **4184** | 0.13 | 0.42 | 0.95 | 0.44 | 4.38 | **0.04** |

| Dataset | Accuracy (↑) | | | | | | Normalized Mutual Information (↑) | | | | | |
|---|---|---|---|---|---|---|---|---|---|---|---|---|
| | PCA | t-SNE | UMAP | AE | MDS | Ψ-DR | PCA | t-SNE | UMAP | AE | MDS | Ψ-DR |
| airway | 0.890 | 0.762 | 0.773 | **0.951** | 0.727 | 0.907 | 0.604 | 0.224 | 0.281 | 0.379 | 0.130 | **0.660** |
| crohn | 0.583 | 0.557 | 0.393 | **0.761** | 0.227 | 0.607 | 0.466 | 272 | 0.220 | **0.511** | 0.068 | 0.490 |
| tonsil | 0.559 | 0.351 | 0.330 | **0.840** | 0.360 | 0.577 | 0.455 | 0.107 | 0.154 | **0.547** | 0.158 | 0.424 |
| jurkat | **1.000** | **1.000** | 0.994 | 0.997 | 0.987 | **1.000** | 0.001 | 0.025 | 0.016 | 0.035 | 0.001 | **0.198** |

## 6.2. Comparison on single-cell expression data

Dimensionality reduction algorithms are widely used in single-cell expression data, which often have very high dimensionality (Tsuyuzaki et al., 2020; Song et al., 2022; Wu & Zhang, 2020; Rebuffet et al., 2024). We compare the algorithms on 4 single-cell expression datasets. The results are shown in Table 5, and the runtime (in CPU seconds) is shown in Table 6. Ψ-DR and PCA achieved the best results on the dimensionality reduction metrics CH and DB, while Ψ-DR and AE performed better on downstream tasks. Similar to the benchmark data, PCA and Ψ-DR were the most efficient algorithms for dimensionality reduction.

*Table 6.* Compute time on single-cell expression datasets.

| Dataset | PCA | t-SNE | UMAP | AE | MDS | Ψ-DR |
|---|---|---|---|---|---|---|
| airway | **0.17** | 13.34 | 49.92 | 21.63 | 224.77 | 1.66 |
| crohn | **0.60** | 109.57 | 108.11 | 106.07 | 6540.80 | 2.35 |
| tonsil | **0.16** | 9.30 | 44.84 | 17.41 | 130.01 | 0.74 |
| jurkat | **0.14** | 2.87 | 16.95 | 8.65 | 11.47 | 0.16 |

## 6.3. Comparison with the latest dimensionality reduction algorithms

Here, we compare our method with state-of-the-art dimensionality reduction algorithms that utilize invariants similar to those used in t-SNE and UMAP. Table 7, 8 and 9 show the results comparing our method with Neg-t-SNE (Damrich et al., 2023), InfoNC-t-SNE (Damrich et al., 2023), PaCMAP (Wang et al., 2021), and LocalMAP (Wang et al., 2025). Experimental results demonstrate that the proposed method achieves superior performance compared to the state-of-the-art approaches in terms of both dimensionality reduction quality and efficiency.

We compared the running times on the Rcv1 dataset across

various random sampling points and dimensions, as shown in Appendix K. The results demonstrate that only Ψ-DR is linear with respect to both the number of points $n$ and $d$.

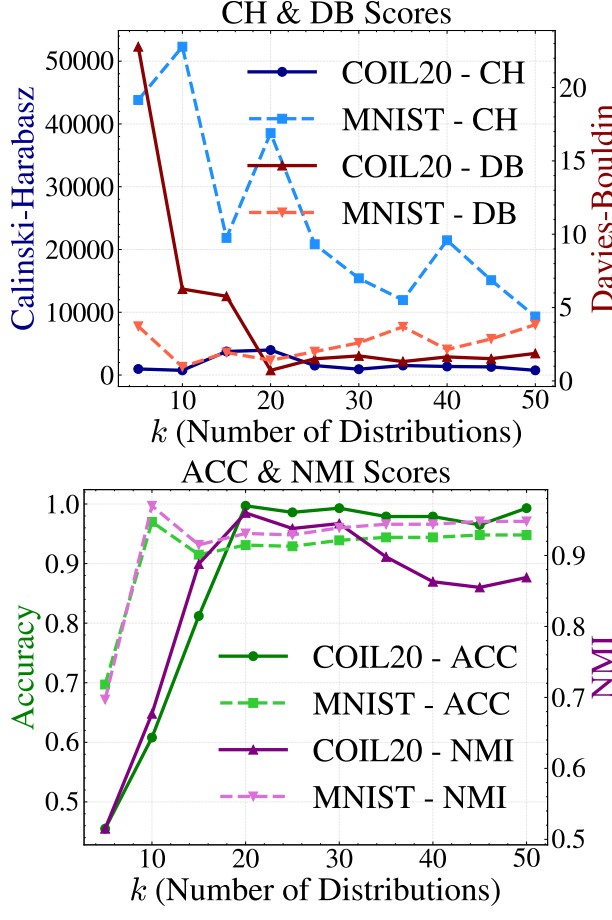

*Figure 2.* Results of dimensionality reduction vary with $k$.

*Table 7.* Dimensionality Reduction Performance Comparison (NtS: Neg-t-SNE, ItS: InfoNC-t-SNE).

| Dataset | Calinski-Harabasz (↑) | | | | | Davies-Bouldin (↓) | | | | |
|---|---|---|---|---|---|---|---|---|---|---|
| | Ψ-DR | NtS | ItS | PaCMAP | LocalMAP | Ψ-DR | NtS | ItS | PaCMAP | LocalMAP |
| COIL20 | 3985 | 626 | 953 | **7758** | 2787 | **0.72** | 3.87 | 4.19 | 3.69 | 4.58 |
| Banknote | **37146** | 552 | 752 | 747 | 714 | **0.05** | 1.40 | 1.21 | 1.10 | 1.21 |
| Landsat | **1990** | 1669 | 1183 | 1892 | 1108 | 1.06 | **1.03** | 1.10 | 1.05 | 1.09 |
| Pendigits | **7470** | 4037 | 3785 | 4357 | 3379 | **1.27** | 3.72 | 2.24 | 2.68 | 4.32 |
| USPS | **8253** | 5012 | 395 | 8192 | 6374 | **1.05** | 1.93 | 1.96 | 1.43 | 1.11 |
| Letters | 419 | **715** | 502 | 541 | 576 | 11.92 | 11.01 | 17.19 | 16.18 | **9.82** |
| MNIST | 52307 | 46188 | 25404 | **81764** | 46269 | 0.96 | 1.48 | 1.76 | 1.29 | **0.87** |
| Skin | **1279417** | 6475 | 3889 | 47859 | 44706 | **0.13** | 3.62 | 4.82 | 1.46 | 1.48 |
| Gisette | **21224** | 6171 | 5549 | 19091 | 9896 | **0.34** | 0.98 | 1.02 | 0.51 | 0.70 |
| Rcv1 | 15722 | NA | NA | **17381** | 13111 | 15.73 | NA | NA | 21.28 | **13.17** |

| Dataset | Accuracy (↑) | | | | | Normalized Mutual Information (↑) | | | | |
|---|---|---|---|---|---|---|---|---|---|---|
| | Ψ-DR | NtS | ItS | PaCMAP | LocalMAP | Ψ-DR | NtS | ItS | PaCMAP | LocalMAP |
| COIL20 | **0.997** | 0.878 | 0.868 | 0.875 | 0.885 | **0.960** | 0.826 | 0.831 | 0.910 | 0.9 |
| Banknote | 0.996 | 0.996 | 0.993 | **1.000** | **1.000** | **0.936** | 0.603 | 0.701 | 0.392 | 0.447 |
| Landsat | 0.855 | 0.848 | 0.853 | 0.855 | **0.870** | 0.681 | 0.633 | 0.576 | **0.690** | 0.643 |
| Pendigits | 0.906 | 0.982 | 0.974 | 0.987 | **0.991** | 0.819 | 0.818 | 0.733 | **0.872** | 0.832 |
| USPS | 0.825 | 0.874 | 0.870 | 0.918 | **0.952** | 0.799 | 0.663 | 0.602 | 0.756 | **0.815** |
| Letters | 0.607 | 0.797 | 0.789 | 0.790 | **0.852** | 0.513 | 0.499 | 0.435 | **0.520** | 0.504 |
| MNIST | **0.970** | 0.876 | 0.853 | 0.919 | 0.933 | **0.814** | 0.630 | 0.600 | 0.740 | 0.807 |
| Skin | 0.996 | 0.998 | 0.989 | 0.998 | **0.999** | **0.828** | 0.086 | 0.047 | 0.022 | 0.035 |
| Gisette | 0.931 | 0.952 | 0.959 | 0.975 | **0.981** | 0.655 | 0.678 | 0.687 | **0.736** | 0.690 |
| Rcv1 | 0.547 | NA | NA | 0.620 | **0.656** | 0.343 | NA | NA | **0.360** | 0.336 |

*Table 8.* Summary of the Comparison.

| | Ψ-DR | NtS | ItS | PaCMAP | LocalMAP |
|---|---|---|---|---|---|
| winner | 18 | 2 | 0 | 9 | 12 |
| runner-up | 11 | 6 | 2 | 17 | 7 |

*Table 9.* Computation Time (seconds).

| Datasets | Ψ-DR | NtS | ItS | PaCMAP | LocalMAP |
|---|---|---|---|---|---|
| COIL20 | **0.66** | 3.72 | 1.35 | 21.62 | 14.09 |
| Banknote | **0.15** | 1.25 | 1.28 | 13.45 | 16.05 |
| Landsat | **0.16** | 1.16 | 2.30 | 14.54 | 13.80 |
| Pendigits | **2.19** | 7.39 | 7.83 | 20.28 | 22.3 |
| USPS | **6.32** | 8.86 | 9.15 | 24.50 | 26.05 |
| Letters | **10.47** | 13.07 | 14.27 | 32.95 | 33.40 |
| MNIST | **28.55** | 66.29 | 70.52 | 87.28 | 82.90 |
| Skin | **22.71** | 174 | 187 | 332 | 339 |
| Gisette | **2.92** | 9.08 | 9.66 | 26.39 | 25.51 |
| Rcv1 | **382** | NA | NA | 2911 | 2809 |

The dimensionality reduction results vary with the number of distributions $k$ are shown in Figure 2. We report the results for different values of $k$ on COIL and MNIST datasets, where COIL contains 20 classes, and MNIST has 10 classes. In terms of all four metrics, Ψ-DR achieved the optimal results near the number of distributions corresponding to the true number of classes in both datasets.

## 7. Conclusion

In this paper, we presented a novel perspective on dimensionality reduction by proposing a distributional invariant that preserves the similarity of points and distributions across the original and reduced spaces. To strictly enforce this invariant, we introduced Ψ-DR (or PSIDR), a linear-time algorithm designed for high-dimensional data. Through rigorous theoretical analysis and extensive experiments on both standard benchmarks and single-cell gene expression datasets, we demonstrated the effectiveness of our approach. Ψ-DR not only yields superior internal validation scores (CH and DB indices) but also significantly enhances performance in downstream tasks such as classification and clustering. Notably, its linear time complexity $O(nd)$ ensures scalability, making Ψ-DR a highly efficient solution for processing massive datasets with hundreds of thousands of samples and tens of thousands of dimensions.

## Acknowledgements

Kai Ming Ting is supported by the National Natural Science Foundation of China (No. W2531050 & 92470116). This project is also supported by Fundamental and Interdisciplinary Disciplines Breakthrough Plan of the Ministry of Education of China (No. JYB2025XDXM118).

## Impact Statement

This paper presents work whose goal is to advance the field of machine learning. There are many potential societal consequences of our work, none of which we feel must be specifically highlighted here.

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

## A. Notations Used

The notations used in the main text are shown in Table 10 and the additional notations used in the proofs are shown in Table 11.

*Table 10.* The notations used and their descriptions.

| Notation | Description |
|---|---|
| $n$ | Number of data points (sample size) |
| $d$ | Dimensionality of the original input space |
| $p$ | Dimensionality of the target embedding space ($p \ll d$) |
| $k$ | Number of underlying distributions (or clusters) |
| $\mathcal{X}$ | Input domain, a compact subset of $\mathbb{R}^d$ |
| $x_i$ | The $i$-th data point, $x_i \in \mathcal{X}$ |
| $\nu_c$ | The $c$-th underlying probability distribution |
| $C_c$ | The set of indices belonging to the $c$-th subset |
| $n_{\min}$ | The size of the smallest subset, $n_{\min} := \min_c |C_c|$ |
| $\kappa(\cdot, \cdot)$ | Continuous, positive definite kernel function |
| $\mathcal{H}_\kappa$ | Reproducing Kernel Hilbert Space (RKHS) induced by $\kappa$ |
| $\phi(\cdot)$ | Feature map $\phi : \mathcal{X} \to \mathcal{H}_\kappa$ |
| $\mu_c$ | True Kernel Mean Embedding (KME) of distribution $\nu_c$ |
| $\hat{\mu}_c$ | Empirical KME estimated from subset $C_c$ |
| $\varepsilon$ | Upper bound of the kernel norm, $\sup_x \kappa(x, x) \le \varepsilon$ |
| $z_i$ | Low-dimensional embedding of $x_i$ in $\mathbb{R}^p$ |
| $\bar{z}_c$ | Low-dimensional embedding (center) of the $c$-th distribution |
| $\mathbf{Z}$ | The embedding matrix of all points, $\mathbf{Z} \in \mathbb{R}^{n \times p}$ |
| $\bar{\mathbf{Z}}$ | The embedding matrix of distributional centers, $\bar{\mathbf{Z}} \in \mathbb{R}^{k \times p}$ |
| $\mathbf{D}$ | Matrix of squared MMD distances between distributions ($k \times k$) |
| $d_{ic}$ | Target Euclidean distance between point $i$ and center $c$ derived from KME |
| $\Delta_p$ | Eigengap between the $p$-th and $(p+1)$-th eigenvalues |
| $\lambda$ | Regularization parameter balancing the two loss terms |

## B. Closed-Form Multilateration via Linearization

The location of each point $z_i \in \mathbb{R}^p$ given the estimated distances $\{d_{ic}\}_{c=1}^k$ to the fixed anchors $\{\bar{z}_c\}_{c=1}^k$ can be formulated as finding the intersection of $k$ hyperspheres. This requires solving the following system of quadratic equations:

$$\|z_i - \bar{z}_c\|_2^2 = d_{ic}^2, \quad c = 1, \ldots, k. \tag{3}$$

Directly minimizing the non-linear least squares objective is computationally expensive and sensitive to initialization. Instead, we employ an efficient **linearized algebraic closed-form solution** (Norrdine, 2012).

Expanding the squared norm yields:

$$\|z_i\|_2^2 - 2z_i^\top \bar{z}_c + \|\bar{z}_c\|_2^2 = d_{ic}^2. \tag{4}$$

Since the term $\|z_i\|_2^2$ is unknown and non-linear, we eliminate it by selecting the first anchor $\bar{z}_1$ as a reference and subtracting its corresponding equation from the equations of the remaining anchors $c = 2, \ldots, k$. The subtraction yields a set of $k - 1$ linear equations:

$$2z_i^\top (\bar{z}_1 - \bar{z}_c) = d_{ic}^2 - d_{i1}^2 - \|\bar{z}_c\|_2^2 + \|\bar{z}_1\|_2^2. \tag{5}$$

This system can be written in the matrix form $\mathbf{A}z_i = \mathbf{b}_i$, where the measurement matrix $\mathbf{A} \in \mathbb{R}^{(k-1) \times p}$ and the observation vector $\mathbf{b}_i \in \mathbb{R}^{k-1}$ are defined as:

$$\mathbf{A}_{c-1} = 2(\bar{z}_1 - \bar{z}_c)^\top, \tag{6}$$

$$(\mathbf{b}_i)_{c-1} = d_{ic}^2 - d_{i1}^2 - \|\bar{z}_c\|_2^2 + \|\bar{z}_1\|_2^2, \tag{7}$$

*Table 11.* Additional notations and mathematical operators used in the proofs.

| Notation | Description |
|---|---|
| $(\cdot)^{\dagger}$ | The Moore-Penrose pseudoinverse of a matrix |
| $\|\cdot\|_F$ | The Frobenius norm of a matrix |
| $\|\cdot\|_{\mathrm{op}}$ | The spectral norm (operator norm) of a matrix |
| $\mathcal{O}(p)$ | The set of $p \times p$ orthogonal matrices |
| $\mathbf{I}_k$ | Identity matrix of size $k \times k$ |
| $\mathbf{H}$ | Centering matrix, $\mathbf{H} = \mathbf{I}_k - \frac{1}{k}\mathbf{1}_k\mathbf{1}_k^{\top}$ |
| $\mathbf{D}^*$ | The "true" squared distance matrix based on population KMEs $\mu_c$ |
| $\mathbf{B}, \mathbf{B}^*$ | Double-centered Gram matrices derived from $\mathbf{D}$ and $\mathbf{D}^*$ |
| $\mathbf{U}_p, \mathbf{U}_p^*$ | Matrices containing the top-$p$ eigenvectors of $\mathbf{B}$ and $\mathbf{B}^*$ |
| $\Theta(\cdot, \cdot)$ | Diagonal matrix of canonical angles between two subspaces |
| $\mathbf{A}$ | The coefficient matrix in the linearized multilateration system |
| $\sigma_0$ | Lower bound on the smallest singular value of $\mathbf{A}$ (conditioning parameter) |
| $\mathbf{b}$ | The observation vector in the linearized multilateration system |
| $z_i^*$ | The theoretical ideal embedding of $x_i$ preserving true Hilbert distances |
| $\mathbf{Q}$ | Optimal orthogonal alignment matrix used in perturbation analysis |
| $L$ | Lipschitz constant for the multilateration solution |
| $\epsilon_n$ | Statistical error bound term, scaling with $O(1/\sqrt{n_{\min}})$ |

for $c = 2, \ldots, k$.

The least-squares solution is then given by the normal equations:

$$z_i = (\mathbf{A}^{\top}\mathbf{A})^{-1}\mathbf{A}^{\top}\mathbf{b}_i. \tag{8}$$

In our implementation, we solve this utilizing singular value decomposition (SVD) for numerical stability.

**Robust Fallback Strategy.** In degenerate configurations where the anchors lie on a lower-dimensional manifold (e.g., collinear anchors in 2D or coplanar in 3D), the matrix $\mathbf{A}$ becomes rank-deficient or ill-conditioned. To handle such cases, we monitor the condition of the linear system. If the linear solver fails (indicating singularity), we fall back to a numerical optimization approach, minimizing the original non-linear objective function via the Broyden–Fletcher–Goldfarb–Shanno (BFGS) algorithm (Head & Zerner, 1985), initialized at the centroid of the anchors.

## C. Proof of Lemma 5.4.

*Lemma* 5.4. Suppose Assumption 5.1 holds. For any $\delta \in (0, 1)$, with probability at least $1 - \delta$,

$$\max_{c \in [k]} \|\hat{\mu}_c - \mu_c\|_{\mathcal{H}_{\kappa}} \leq \sqrt{\frac{\varepsilon}{n_{\min}}}\left(1 + \sqrt{6 \log \frac{2k}{\delta}}\right).$$

*Proof.* Fix an arbitrary subset $c \in [k]$, and let $n_c = |C_c|$. The empirical kernel mean embedding is given by

$$\hat{\mu}_c = \frac{1}{n_c}\sum_{i \in C_c}\phi(x_i),$$

where the samples $\{x_i\}_{i \in C_c}$ are assumed to be i.i.d. draws from the distribution $\nu_c$. By Assumption 5.1, the kernel is bounded, which implies $\|\phi(x)\|_{\mathcal{H}_{\kappa}}^2 = k(x, x) \leq \varepsilon$ for all $x \in \mathcal{X}$. Consequently, each summand $\phi(x_i)$ is a bounded random variable in the Hilbert space $\mathcal{H}_{\kappa}$ with $\|\phi(x_i)\|_{\mathcal{H}_{\kappa}} \leq \sqrt{\varepsilon}$ almost surely.

We apply the upper tail of Talagrand's inequality, Bousquet's version (see, e.g., Theorem 3.3.9 in Giné & Nickl (2021)). For any $t > 0$, let

$$Z = n_c\|\hat{\mu}_c - \mu_c\|_{\mathcal{H}_{\kappa}} = \sup_{f \in \mathcal{H}_{\kappa}, \|f\| \leq 1}\sum_{i=1}^{n_c}\left(f(x_i) - \mathbb{E}[f(x)]\right),$$

for $f$, $\|f\|_{\mathcal{H}_\kappa} \leq 1$, we have $|f(x)| \leq \sqrt{\kappa(x,x)} \leq \sqrt{\varepsilon}$, so:

$$|f(x) - \mathbb{E}[f(x)]| \leq |f(x)| + |\mathbb{E}[f(x)]| \leq \sqrt{\varepsilon} + \sqrt{\varepsilon} = 2\sqrt{\varepsilon} = U.$$

$$\sup_{f \in \mathcal{H}_\kappa, \|f\| \leq 1} \mathbb{E}\left[(f(x))^2\right] \leq \sup_f \|f\|_\infty^2 \leq \varepsilon = \sigma^2$$

$$\mathbb{E}[Z] = n_c \mathbb{E}\|\hat{\mu}_c - \mu_c\| \leq n_c \sqrt{\mathbb{E}\|\hat{\mu}_c - \mu_c\|^2} \leq n_c \sqrt{\frac{\varepsilon}{n_c}} = \sqrt{n_c \varepsilon}$$

$$V = n_c \sigma^2 + 2U\mathbb{E}[Z] \leq n_c \varepsilon + 2(2\sqrt{\varepsilon})(\sqrt{n_c \varepsilon}) = n_c \varepsilon + 4\varepsilon\sqrt{n_c} \leq 2n_c \varepsilon. (n_c \geq 16)$$

this yields

$$\mathbb{P}\left(\|\hat{\mu}_c - \mu_c\|_{\mathcal{H}_\kappa} \geq \sqrt{\frac{\varepsilon}{n_c}} + t\right) \leq \exp\left(-\frac{n_c t^2}{4\varepsilon + \frac{4}{3}\sqrt{\varepsilon}t}\right).$$

For $t \leq \sqrt{\varepsilon}$, the denominator is at most $6\varepsilon$, so the bound simplifies to

$$\mathbb{P}\left(\|\hat{\mu}_c - \mu_c\|_{\mathcal{H}_\kappa} \geq \sqrt{\frac{\varepsilon}{n_c}} + t\right) \leq \exp\left(-\frac{n_c t^2}{6\varepsilon}\right).$$

To obtain a high-probability bound, we set the right-hand side equal to $\delta/(2k)$. Solving for $t$ gives

$$t = \sqrt{\frac{6\varepsilon}{n_c} \log \frac{2k}{\delta}}.$$

Thus, with probability at least $1 - \delta/(2k)$,

$$\|\hat{\mu}_c - \mu_c\|_{\mathcal{H}_\kappa} \leq \sqrt{\frac{\varepsilon}{n_c}} + \sqrt{\frac{6\varepsilon}{n_c} \log \frac{2k}{\delta}} = \sqrt{\frac{\varepsilon}{n_c}}\left(1 + \sqrt{6 \log \frac{2k}{\delta}}\right).$$

Since $n_c \geq n_{\min}$ by definition of $n_{\min}$, and the function $x \mapsto 1/\sqrt{x}$ is decreasing, we have $\sqrt{1/n_c} \leq \sqrt{1/n_{\min}}$. Therefore, the bound holds with $n_{\min}$ in place of $n_c$:

$$\|\hat{\mu}_c - \mu_c\|_{\mathcal{H}_\kappa} \leq \sqrt{\frac{\varepsilon}{n_{\min}}}\left(1 + \sqrt{6 \log \frac{2k}{\delta}}\right).$$

This bound holds for a fixed subset $c$. To obtain a uniform bound over all $k$ subsets, we apply the union bound. Specifically, let $E_c$ denote the event that the above inequality fails for subset $c$. Then $\mathbb{P}(E_c) \leq \delta/(2k)$, and

$$\mathbb{P}\left(\bigcup_{c=1}^k E_c\right) \leq \sum_{c=1}^k \mathbb{P}(E_c) \leq k \cdot \frac{\delta}{2k} = \frac{\delta}{2} < \delta.$$

Thus, with probability at least $1 - \delta$, the inequality holds simultaneously for all $c \in [k]$, which implies

$$\max_{c \in [k]} \|\hat{\mu}_c - \mu_c\|_{\mathcal{H}_\kappa} \leq \sqrt{\frac{\varepsilon}{n_{\min}}}\left(1 + \sqrt{6 \log \frac{2k}{\delta}}\right).$$

$\square$

# D. Proof of Lemma 5.5.

*Lemma* 5.5. Let $\mathbf{D}, \mathbf{D}^* \in \mathbb{R}^{k \times k}$ be Euclidean distance matrices, and let $\bar{\mathbf{Z}}, \bar{\mathbf{Z}}^* \in \mathbb{R}^{k \times p}$ be their classical MDS embeddings (i.e., $\bar{\mathbf{Z}}\bar{\mathbf{Z}}^\top = \mathbf{B}$, where $\mathbf{B} = -\frac{1}{2}\mathbf{H}\mathbf{D}\mathbf{H}$, and similarly for $\bar{\mathbf{Z}}^*$). Let $\mathcal{O}(p)$ denote the set of $p \times p$ orthogonal matrices. Under Assumption 3,

$$\min_{\mathbf{Q} \in \mathcal{O}(p)} \left\| \bar{\mathbf{Z}} - \bar{\mathbf{Z}}^* \mathbf{Q} \right\|_F \leq \frac{\sqrt{2p}}{2\Delta_p} \left\| \mathbf{D} - \mathbf{D}^* \right\|_F.$$

*Proof.* Let $\mathbf{H} = \mathbf{I}_k - \frac{1}{k}\mathbf{1}_k\mathbf{1}_k^\top$ be the centering matrix. The Gram matrices corresponding to $\mathbf{D}$ and $\mathbf{D}^*$ are

$$\mathbf{B} = -\frac{1}{2}\mathbf{H}\mathbf{D}\mathbf{H}, \quad \mathbf{B}^* = -\frac{1}{2}\mathbf{H}\mathbf{D}^*\mathbf{H}.$$

Their difference satisfies

$$\mathbf{B} - \mathbf{B}^* = -\frac{1}{2}\mathbf{H}(\mathbf{D} - \mathbf{D}^*)\mathbf{H}.$$

Since $\mathbf{H}$ is symmetric and idempotent ($\mathbf{H}^\top = \mathbf{H}$, $\mathbf{H}^2 = \mathbf{H}$), its spectral norm is $\|\mathbf{H}\|_2 = 1$. Using the submultiplicativity of the Frobenius norm with respect to the spectral norm, we obtain

$$\|\mathbf{B} - \mathbf{B}^*\|_F = \frac{1}{2}\|\mathbf{H}(\mathbf{D} - \mathbf{D}^*)\mathbf{H}\|_F \leq \frac{1}{2}\|\mathbf{H}\|_2\|\mathbf{D} - \mathbf{D}^*\|_F\|\mathbf{H}\|_2 = \frac{1}{2}\|\mathbf{D} - \mathbf{D}^*\|_F.$$

Let $\mathbf{U}_p, \mathbf{U}_p^* \in \mathbb{R}^{k \times p}$ be the matrices whose columns are the top-$p$ eigenvectors of $\mathbf{B}$ and $\mathbf{B}^*$, respectively. Under Assumption 3, the eigengap $\Delta_p = \lambda_p^* - \lambda_{p+1}^*$ is positive. By the Davis–Kahan $\sin\Theta$ theorem (Bhatia (1997) Theorem VII.3.1), the Frobenius norm of the difference between the orthogonal projectors onto the top-$p$ eigenspaces satisfies

$$\|\mathbf{U}_p\mathbf{U}_p^\top - \mathbf{U}_p^*\mathbf{U}_p^{*\top}\|_F = \|\sin\Theta(\mathbf{U}_p, \mathbf{U}_p^*)\|_F \leq \cdot\sqrt{p}\,\|\sin\Theta(\mathbf{U}_p, \mathbf{U}_p^*)\|_{op} \leq \sqrt{p}\,\frac{\|\mathbf{B} - \mathbf{B}^*\|_{op}}{\Delta_p}.$$

Finally, using the general norm inequality $\|\mathbf{B} - \mathbf{B}^*\|_{op} \leq \|\mathbf{B} - \mathbf{B}^*\|_F$, we obtain the stated result:

$$\|\mathbf{U}_p\mathbf{U}_p^\top - \mathbf{U}_p^*\mathbf{U}_p^{*\top}\|_F \leq \frac{\sqrt{p}\,\|\mathbf{B} - \mathbf{B}^*\|_F}{\Delta_p}.$$

The classical MDS embeddings are defined as $\bar{\mathbf{Z}} = \mathbf{U}_p\mathbf{\Lambda}_p^{1/2}$ and $\bar{\mathbf{Z}}^* = \mathbf{U}_p^*\mathbf{\Lambda}_p^{*1/2}$, where $\mathbf{\Lambda}_p, \mathbf{\Lambda}_p^*$ are the diagonal matrices of the top-$p$ eigenvalues. By Theorem 2 of Yu et al. (2015), which bounds the perturbation of low-rank factorizations under eigengap conditions, we have

$$\min_{\mathbf{Q} \in \mathcal{O}(p)} \left\| \bar{\mathbf{Z}} - \bar{\mathbf{Z}}^* \mathbf{Q} \right\|_F \leq \sqrt{2}\|\mathbf{U}_p\mathbf{U}_p^\top - \mathbf{U}_p^*\mathbf{U}_p^{*\top}\|_F \leq \frac{\sqrt{2p}}{\Delta_p}\|\mathbf{B} - \mathbf{B}^*\|_F.$$

Substituting the bound $\|\mathbf{B} - \mathbf{B}^*\|_F \leq \frac{1}{2}\|\mathbf{D} - \mathbf{D}^*\|_F$ into the above inequality yields

$$\min_{\mathbf{Q} \in \mathcal{O}(p)} \left\| \bar{\mathbf{Z}} - \bar{\mathbf{Z}}^* \mathbf{Q} \right\|_F \leq \frac{\sqrt{2p}}{2\Delta_p}\|\mathbf{D} - \mathbf{D}^*\|_F.$$

$\square$

# E. Proof of Lemma 5.6.

*Lemma* 5.6. Let $\bar{\mathbf{Z}} = [\bar{z}_1, \ldots, \bar{z}_k]^\top \in \mathbb{R}^{k \times p}$ have full column rank, and define $\mathbf{A} = -2(\bar{\mathbf{Z}}_{2:k} - \mathbf{1}_{k-1}\bar{z}_1^\top) \in \mathbb{R}^{(k-1) \times p}$. Assume $\sigma_{\min}(\mathbf{A}) \geq \sigma_0 > 0$. Let $\mathbf{d} \in \mathbb{R}^k$ be distances, and define $\mathbf{b} \in \mathbb{R}^{k-1}$ by $b_c = d_{c+1}^2 - d_1^2 - (\|\bar{z}_{c+1}\|_2^2 - \|\bar{z}_1\|_2^2)$. Let $z^\dagger = \mathbf{A}^\dagger \mathbf{b}$. Then for any two inputs $(\mathbf{d}, \bar{\mathbf{Z}})$ and $(\mathbf{d}', \bar{\mathbf{Z}}')$,

$$\|z^\dagger - z^{\dagger'}\|_2 \leq L \left( \|\mathbf{d}^2 - \mathbf{d}'^2\|_2 + \|\bar{\mathbf{Z}} - \bar{\mathbf{Z}}'\|_F \right),$$

where the Lipschitz constant satisfies $L = O\left( \frac{1}{\sigma_0^2} \left( 1 + \frac{|\mathbf{b}|_2}{\sigma_0} \right) \right)$.

*Proof.* The multilateration solution $z^\dagger$ satisfies the normal equations $\mathbf{A}^\top \mathbf{A} z^\dagger = \mathbf{A}^\top \mathbf{b}$. Consider perturbed inputs $(\mathbf{d}', \bar{\mathbf{Z}}')$ yielding $\mathbf{A}'$ and $\mathbf{b}'$, with solution $z^{\dagger'}$ satisfying $\mathbf{A}'^\top \mathbf{A}' z^{\dagger'} = \mathbf{A}'^\top \mathbf{b}'$.

Subtracting the two equations gives

$$\mathbf{A}^\top \mathbf{A}(z^\dagger - z^{\dagger'}) = \mathbf{A}'^\top \mathbf{b}' - \mathbf{A}^\top \mathbf{b} - \left( \mathbf{A}'^\top \mathbf{A}' - \mathbf{A}^\top \mathbf{A} \right) z^{\dagger'}.$$

Let $\Delta \mathbf{A} = \mathbf{A}' - \mathbf{A}$ and $\Delta \mathbf{b} = \mathbf{b}' - \mathbf{b}$. Then

$$\mathbf{A}'^\top \mathbf{b}' - \mathbf{A}^\top \mathbf{b} = \mathbf{A}^\top \Delta \mathbf{b} + \Delta \mathbf{A}^\top \mathbf{b} + \Delta \mathbf{A}^\top \Delta \mathbf{b},$$

$$\mathbf{A}'^\top \mathbf{A}' - \mathbf{A}^\top \mathbf{A} = \mathbf{A}^\top \Delta \mathbf{A} + \Delta \mathbf{A}^\top \mathbf{A} + \Delta \mathbf{A}^\top \Delta \mathbf{A}.$$

For sufficiently small perturbations, higher-order terms $\Delta \mathbf{A}^\top \Delta \mathbf{b}$ and $\Delta \mathbf{A}^\top \Delta \mathbf{A}$ are negligible. Keeping only first-order terms,

$$\mathbf{A}^\top \mathbf{A}(z^\dagger - z^{\dagger'}) \approx \mathbf{A}^\top \Delta \mathbf{b} + \Delta \mathbf{A}^\top \mathbf{b} - (\mathbf{A}^\top \Delta \mathbf{A} + \Delta \mathbf{A}^\top \mathbf{A}) z^{\dagger'}.$$

Taking norms and using $\|\mathbf{A}^\top \mathbf{A}\|_2 \geq \sigma_0^2$,

$$\sigma_0^2 \|z^\dagger - z^{\dagger'}\|_2 \leq \|\mathbf{A}\|_2 \|\Delta \mathbf{b}\|_2 + \|\Delta \mathbf{A}\|_2 \|\mathbf{b}\|_2 + 2\|\mathbf{A}\|_2 \|\Delta \mathbf{A}\|_2 \|z^{\dagger'}\|_2. \tag{9}$$

We now derive explicit bounds for each term on the right-hand side.

Recall that $\mathbf{A} = -2\mathbf{L}\bar{\mathbf{Z}}$, where $\mathbf{L}$ is a differencing matrix with spectral norm $\|\mathbf{L}\|_2 = \sqrt{k}$. Thus,

$$\|\mathbf{A}\|_2 \leq 2\|\mathbf{L}\|_2 \|\bar{\mathbf{Z}}\|_2 = 2\sqrt{k}\|\bar{\mathbf{Z}}\|_2.$$

By linearity, $\Delta \mathbf{A} = -2\mathbf{L}(\bar{\mathbf{Z}}' - \bar{\mathbf{Z}})$. Using the fact that the spectral norm is bounded by the Frobenius norm, we have

$$\|\Delta \mathbf{A}\|_2 \leq 2\sqrt{k}\|\bar{\mathbf{Z}}' - \bar{\mathbf{Z}}\|_2 \leq 2\sqrt{k}\|\bar{\mathbf{Z}}' - \bar{\mathbf{Z}}\|_F.$$

The vector $\mathbf{b}$ is defined component-wise as $b_c = (d_{c+1}^2 - d_1^2) - (\|\bar{z}_{c+1}\|_2^2 - \|\bar{z}_1\|_2^2)$. The perturbation $\Delta \mathbf{b}$ splits into a distance part and an anchor coordinate part:

$$\|\Delta \mathbf{b}\|_2 \leq \|\Delta \mathbf{b}_{\text{dist}}\|_2 + \|\Delta \mathbf{b}_{\text{coord}}\|_2.$$

For the distance part, each component is a difference of two squared distances. The norm is bounded by:

$$\|\Delta \mathbf{b}_{\text{dist}}\|_2 \leq 2\|\mathbf{d}^2 - \mathbf{d}'^2\|_2.$$

For the coordinate part, the function $f(z) = \|z\|^2$ is locally Lipschitz with constant $2\|z\|$. The term $\|\bar{z}_{c+1}\|^2 - \|\bar{z}_1\|^2$ involves two such terms, leading to a Lipschitz constant bounded by $4 \max_c \|\bar{z}_c\|_2$. Thus:

$$\|\Delta \mathbf{b}_{\text{coord}}\|_2 \leq 4 \max_c \|\bar{z}_c\|_2 \|\bar{\mathbf{Z}} - \bar{\mathbf{Z}}'\|_F.$$

Combining these, we obtain:

$$\|\Delta \mathbf{b}\|_2 \leq 2\|\mathbf{d}^2 - \mathbf{d}'^2\|_2 + 4 \max_c \|\bar{z}_c\|_2 \|\bar{\mathbf{Z}} - \bar{\mathbf{Z}}'\|_F.$$

Substituting the bounds from Steps 1 and 2, and using $\|z^{\dagger'}\|_2 \lesssim \|\mathbf{b}\|_2/\sigma_0$ ($\|z^{\dagger'}\|_2 \leq \|\mathbf{A}^{\dagger}\|_2\|\mathbf{b}'\|_2$ neglecting higher-order terms), into (9):

$$\sigma_0^2\|z^{\dagger} - z^{\dagger'}\|_2 \leq \underbrace{\left(2\sqrt{k}\|\bar{\mathbf{Z}}\|_2\right)}_{\|\mathbf{A}\|_2}\left(2\|\mathbf{d}^2 - \mathbf{d}'^2\|_2 + 4\max_c\|\bar{z}_c\|_2\|\bar{\mathbf{Z}} - \bar{\mathbf{Z}}'\|_F\right)$$

$$+ \underbrace{\left(2\sqrt{k}\|\bar{\mathbf{Z}} - \bar{\mathbf{Z}}'\|_F\right)}_{\|\Delta\mathbf{A}\|_2}\|\mathbf{b}\|_2$$

$$+ 2\underbrace{\left(2\sqrt{k}\|\bar{\mathbf{Z}}\|_2\right)}_{\|\mathbf{A}\|_2}\underbrace{\left(2\sqrt{k}\|\bar{\mathbf{Z}} - \bar{\mathbf{Z}}'\|_F\right)}_{\|\Delta\mathbf{A}\|_2}\frac{\|\mathbf{b}\|_2}{\sigma_0}.$$

Now, we strictly group the terms by the perturbation sources $\|\mathbf{d}^2 - \mathbf{d}'^2\|_2$ and $\|\bar{\mathbf{Z}} - \bar{\mathbf{Z}}'\|_F$.

The coefficient for $\|\mathbf{d}^2 - \mathbf{d}'^2\|_2$ is:

$$C_1 = 4\sqrt{k}\|\bar{\mathbf{Z}}\|_2.$$

The coefficient for $\|\bar{\mathbf{Z}} - \bar{\mathbf{Z}}'\|_F$ is sum of three terms:

$$C_2 = \left(8\sqrt{k}\|\bar{\mathbf{Z}}\|_2\max_c\|\bar{z}_c\|_2\right) + \left(2\sqrt{k}\|\mathbf{b}\|_2\right) + \left(\frac{8k\|\bar{\mathbf{Z}}\|_2\|\mathbf{b}\|_2}{\sigma_0}\right).$$

Dividing the entire inequality by $\sigma_0^2$, we get:

$$\|z^{\dagger} - z^{\dagger'}\|_2 \leq \frac{C_1}{\sigma_0^2}\|\mathbf{d}^2 - \mathbf{d}'^2\|_2 + \frac{C_2}{\sigma_0^2}\|\bar{\mathbf{Z}} - \bar{\mathbf{Z}}'\|_F.$$

Finally, define the global Lipschitz constant $L$ as the maximum of the two coefficients:

$$L = \max\left(\frac{C_1}{\sigma_0^2}, \frac{C_2}{\sigma_0^2}\right) = O\left(\frac{1}{\sigma_0^2}\left(1 + \frac{|\mathbf{b}|_2}{\sigma_0}\right)\right).$$

This yields the desired form:

$$\|z^{\dagger} - z^{\dagger'}\|_2 \leq L\left(\|\mathbf{d}^2 - \mathbf{d}'^2\|_2 + \|\bar{\mathbf{Z}} - \bar{\mathbf{Z}}'\|_F\right).$$

$\square$

## F. Proof of Theorem 5.7.

*Theorem* 5.7. Let $z_i^* \in \mathbb{R}^p$ satisfy $\|z_i^* - \bar{z}_c^*\|_2^2 = 2(1 - \langle\phi(x_i), \mu_c\rangle_{\mathcal{H}_\kappa})$ for all $c$. Let $z_i$ be the embedding from Algorithm 1. Then, as $n \to \infty$,

$$\frac{1}{n}\sum_{i=1}^{n}\|z_i - z_i^*\|_2^2 \xrightarrow{\mathbb{P}} 0.$$

*Proof.* By Lemma 5.4, with probability $\geq 1 - \delta$, $\max_c\|\hat{\mu}_c - \mu_c\|_{\mathcal{H}_\kappa} \leq \epsilon_n := \sqrt{\frac{\varepsilon}{n_{\min}}}(1 + \sqrt{6\log\frac{2k}{\delta}})$.

Let $d_{ic}^2 = \|\phi(x_i) - \hat{\mu}_c\|_{\mathcal{H}_\kappa}^2$ and $(d_{ic}^*)^2 = \|\phi(x_i) - \mu_c\|_{\mathcal{H}_\kappa}^2$.

Using the algebraic identity $|a^2 - b^2| = |a - b| \cdot |a + b|$:

$$\left|d_{ic}^2 - (d_{ic}^*)^2\right| \leq \underbrace{\|(\phi(x_i) - \hat{\mu}_c) - (\phi(x_i) - \mu_c)\|_{\mathcal{H}_\kappa}}_{\text{Error term}} \cdot \underbrace{(\|\phi(x_i) - \hat{\mu}_c\|_{\mathcal{H}_\kappa} + \|\phi(x_i) - \mu_c\|_{\mathcal{H}_\kappa})}_{\text{Scale term}}.$$

The error term simplifies to $\|\hat{\mu}_c - \mu_c\|_{\mathcal{H}_\kappa} \leq \epsilon_n$. The scale term is bounded by $2\sqrt{\varepsilon} + 2\sqrt{\varepsilon} = 4\sqrt{\varepsilon}$. Thus, $|d_{ic}^2 - d_{ic}^{*\,2}| \leq 4\sqrt{\varepsilon}\epsilon_n$.

$$|\mathbf{d}_i^2 - (\mathbf{d}_i^*)^2|2 = \sqrt{\sum c = 1^k |d_{ic}^2 - (d_{ic}^*)^2|^2} \le \sqrt{k \cdot (4\sqrt{\varepsilon}\epsilon_n)^2} = 4\epsilon_n\sqrt{k\varepsilon}. \tag{10}$$

Similarly, for the MMD matrix, let $D_{cc'} = \|\hat{\mu}_c - \hat{\mu}_{c'}\|^2$ and $D_{cc'}^* = \|\mu_c - \mu_{c'}\|^2$. The error term is bounded by $\|\hat{\mu}_c - \mu_c\| + \|\hat{\mu}_{c'} - \mu_{c'}\| \le 2\epsilon_n$. The scale term is bounded by $4\sqrt{\varepsilon}$. Thus, $|D_{cc'} - D_{cc'}^*| \le (2\epsilon_n)(4\sqrt{\varepsilon}) = 8\sqrt{\varepsilon}\epsilon_n$.

Converting to the Frobenius norm for the $k \times k$ matrix:

$$|\mathbf{D} - \mathbf{D}^*|F = \sqrt{\sum c, c'|D_{cc'} - D_{cc'}^*|^2} \le \sqrt{k^2 \cdot (8\sqrt{\varepsilon}\epsilon_n)^2} = k \cdot 8\sqrt{\varepsilon}\epsilon_n. \tag{11}$$

By Lemma 5.5, $\min_{\mathbf{Q}} \|\bar{\mathbf{Z}} - \bar{\mathbf{Z}}^*\mathbf{Q}\|_F \le C_1 k\epsilon_n$ for $C_1 = 4\sqrt{2p\varepsilon}/\Delta_p$.

Lemma 5.6 then gives
$$\|z_i - z_i^*\mathbf{Q}\|_2 \le L\left(\|\mathbf{d}_i^2 - (\mathbf{d}_i^*)^2\|_2 + \|\bar{\mathbf{Z}} - \bar{\mathbf{Z}}^*\mathbf{Q}\|_F\right).$$

So:

$$\|z_i - z_i^*\mathbf{Q}\|_2 \le L\left(\underbrace{\sqrt{k}C_2\epsilon_n}_{\text{Distance term}} + \underbrace{C_1 k\epsilon_n}_{\text{Anchor term}}\right),$$

where $C_2 = 4\sqrt{\varepsilon}$.

Let $C_{\text{total}} = L(C_2 + C_1)$. Thus: $\|z_i - z_i^*\mathbf{Q}\|_2 \le C_{\text{total}} \cdot k\epsilon_n$.

Finally, squaring the last inequality and averaging over all $n$ points yield:

$$\frac{1}{n}\sum_{i=1}^n \|z_i - z_i^*\mathbf{Q}\|_2^2 \le C_{\text{total}}^2 k^2\epsilon_n^2.$$

Recall that $\epsilon_n = O(1/\sqrt{n_{\min}})$; and under Assumption 5.2, $n_{\min} \to \infty$ as $n \to \infty$. Therefore, $\epsilon_n \to 0$, which implies that the mean squared error converges to 0 in probability. $\square$

## G. Proof of Theorem 5.8.

*Theorem* 5.8. Under Assumptions 5.1, 5.2, and 5.3, there exists a constant $C > 0$, such that for any $\delta \in (0, 1)$, with probability at least $1 - \delta$,
$$\frac{1}{n}\sum_{i=1}^n \|z_i - z_i^*\|_2^2 \le C \cdot \frac{k^2}{n_{\min}}\left(1 + \log\frac{2k}{\delta}\right).$$

*Proof.* From the proof of Theorem 5.7, there exists a constant $C_{\text{total}} > 0$ such that for all $i$, with probability $\ge 1 - \delta$,

$$\|z_i - z_i^*\mathbf{Q}\|_2 \le C_{\text{total}} k\epsilon_n, \quad \text{where} \quad \epsilon_n = \sqrt{\frac{\varepsilon}{n_{\min}}}\left(1 + \sqrt{6\log\frac{2k}{\delta}}\right).$$

Squaring both sides and averaging over $i$ give

$$\frac{1}{n}\sum_{i=1}^n \|z_i - z_i^*\|_2^2 = \frac{1}{n}\sum_{i=1}^n \|z_i - z_i^*\mathbf{Q}\|_2^2 \le C_{\text{total}}^2 k^2\epsilon_n^2.$$

Using the inequality $(1 + \sqrt{a})^2 \le 2(1 + a)$ for $a \ge 0$, we obtain

$$\epsilon_n^2 \le \frac{2\varepsilon}{n_{\min}}\left(1 + 6\log\frac{2k}{\delta}\right) \le \frac{12}{n_{\min}}\left(1 + \log\frac{2k}{\delta}\right).$$

Setting $C = 12\varepsilon C_{\text{total}}^2$ completes the proof. $\square$

# H. Proof of Theorem 5.9.

*Theorem* 5.9. Assume the kernel is normalized ($k(x, x) = 1$ for all $x$). Let $d_{ic}^{\mathcal{H}} = \|\phi(x_i) - \mu_c\|_{\mathcal{H}_\kappa}$ and $d_{ic}^{\mathbb{R}^p} = \|z_i - \bar{z}_c\|_2$. Under Assumptions 5.1, 5.2, and 5.3, for any $\delta \in (0, 1)$, with probability at least $1 - \delta$,

$$\max_{i \in [n], c \in [k]} \left| d_{ic}^{\mathbb{R}^p} - d_{ic}^{\mathcal{H}} \right| \le C' \cdot \frac{k}{\sqrt{n_{\min}}} \left( 1 + \sqrt{\log \frac{2k}{\delta}} \right),$$

for some constant $C' > 0$.

*Proof.* We aim to bound the difference between the recovered Euclidean distance and the true Hilbert space distance:

$$\Delta_{ic} = \left| d_{ic}^{\mathbb{R}^p} - d_{ic}^{\mathcal{H}} \right| = \left| \|z_i - \bar{z}_c\|_2 - \|\phi(x_i) - \mu_c\|_{\mathcal{H}_\kappa} \right|.$$

Recall from Theorem 5.7 that the "true" embeddings $z_i^*$ and centers $\bar{z}_c^*$ are defined to perfectly preserve the Hilbert space distances. That is, for any rotation matrix $\mathbf{Q} \in \mathcal{O}(p)$,

$$\|\phi(x_i) - \mu_c\|_{\mathcal{H}_\kappa} = \|z_i^* - \bar{z}_c^*\|_2 = \|z_i^* \mathbf{Q} - \bar{z}_c^* \mathbf{Q}\|_2.$$

Substituting this into the expression for $\Delta_{ic}$:

$$\Delta_{ic} = \left| \|z_i - \bar{z}_c\|_2 - \|z_i^* \mathbf{Q} - \bar{z}_c^* \mathbf{Q}\|_2 \right|.$$

Using the reverse triangle inequality $|\|u\| - \|v\|| \le \|u - v\|$, we have:

$$\Delta_{ic} \le \|(z_i - \bar{z}_c) - (z_i^* \mathbf{Q} - \bar{z}_c^* \mathbf{Q})\|_2.$$

By the standard triangle inequality, we can separate the error contributions from the point $z_i$ and the center $\bar{z}_c$:

$$\Delta_{ic} \le \|z_i - z_i^* \mathbf{Q}\|_2 + \|\bar{z}_c - \bar{z}_c^* \mathbf{Q}\|_2.$$

We now bound each term using our previous results. Let $\mathbf{Q}$ be the optimal alignment matrix from Lemma 5.5.

From Lemma 5.5 and the derivation in Theorem 5.7, the MDS centers satisfy:

$$\|\bar{z}_c - \bar{z}_c^* \mathbf{Q}\|_2 \le \|\bar{\mathbf{Z}} - \bar{\mathbf{Z}}^* \mathbf{Q}\|_F \le C_1 k \epsilon_n.$$

From the proof of Theorem 5.7, the multilateration solution satisfies:

$$\|z_i - z_i^* \mathbf{Q}\|_2 \le C_{\text{total}} k \epsilon_n.$$

Summing these bounds:

$$\Delta_{ic} \le (C_{\text{total}} + C_1) k \epsilon_n.$$

Recall that $\epsilon_n = \sqrt{\frac{\varepsilon}{n_{\min}}} \left( 1 + \sqrt{6 \log \frac{2k}{\delta}} \right)$. Defining a new constant $C' = C_{\text{total}} + C_1$, and noting that $1 + \sqrt{6X} \le \sqrt{6}(1 + \sqrt{X})$ allows us to absorb constants for a cleaner form, we obtain:

$$\max_{i,c} \left| d_{ic}^{\mathbb{R}^p} - d_{ic}^{\mathcal{H}} \right| \le C' \cdot \frac{k}{\sqrt{n_{\min}}} \left( 1 + \sqrt{\log \frac{2k}{\delta}} \right).$$

$\square$

## I. Limitation

The performance of $\Psi$-DR is determined by the 'weakest link', the least represented distribution. To achieve a low embedding error, the dataset should ideally be balanced; for highly imbalanced data, $n_{min}$ becomes the bottleneck.

## J. Experimental Setting

The characteristics of the datasets used are shown in Table 12. Rcv1 is a multi-label dataset, and only the first label of each point is used.

*Table 12.* Characteristics of the datasets used.

| Datasets | #points | #dimensions | $k$ |
|---|---|---|---|
| COIL20 | 1440 | 1024 | 20 |
| Banknote | 1372 | 4 | 2 |
| Landsat | 2000 | 36 | 6 |
| Pendigits | 10992 | 16 | 10 |
| USPS | 11000 | 256 | 10 |
| Letters | 20000 | 16 | 26 |
| MNIST | 70000 | 784 | 10 |
| Skin | 245057 | 3 | 2 |
| Gisette | 7000 | 5000 | 2 |
| Rcv1 | 804414 | 47236 | 37 |

The characteristics of the single-cell expression datasets used are shown in Table 13. We use the highly variable gene (HVG) selection method (Zhao et al., 2025) to select 2000 genes before dimensionality reduction. We set $k$ to be the number of types, and Isolation kernel (Ting et al., 2018) was used in KBC in our experiments.

*Table 13.* Characteristics of the single-cell expression datasets used.

| Datasets | #Cells | #Genes | #Types | Accession | Description |
|---|---|---|---|---|---|
| airway | 7193 | 27716 | 7 | GSE103354 | mouse tracheal epithelium cells |
| crohn | 39563 | 33660 | 27 | Broad Institute Single Cell Portal | human uninflamed ileum |
| tonsil | 5778 | 36601 | 13 | Broad Institute Single Cell Portal | human tonsil |
| jurkat | 3234 | 32738 | 2 | 10X genomics | artificially planted cells |

## K. Scale-up test

The results of the runtime are shown in Table 14. Our method is linear with respect to both the number of points $n$ and the number of dimensions $d$, while Neg-t-SNE and InfoNC-t-SNE are linear with respect to $d$ but have greater than linear time complexity with respect to $n$. PaCMAP and LocalMAP also have greater than linear time complexity with respect to $n$, but they are less affected by $d$.

## L. Ablation Study

Our contribution is proposing a point-distribution similarity invariant, so we conducted the following ablation experiments. We compared the results of maintaining this invariant ($\Psi$-DR) with those of maintaining the point-point similarity invariant (PP-DR). The point-point similarity invariant reduces dimensionality by preserving the distance between each point in $\{C_c\}_{c=1}^k$ (the second line of the algorithm) and the center of $\{C_c\}_{c=1}^k$. The results are shown in Table 15. The results demonstrate that our proposed point-distribution similarity invariant is effective.

*Table 14.* Runtime and Scale-up test.

| | | Time | | | | | Ratio | | | | | |
|---|---|---|---|---|---|---|---|---|---|---|---|---|
| | | $\Psi$-DR | NtS | ItS | PaCMAP | LocalMAP | | $\Psi$-DR | NtS | ItS | PaCMAP | LocalMAP |
| $n$ | 10000 | 5.84 | 51.23 | 36.32 | 11.32 | 11.38 | 1 | 1.00 | 1.00 | 1.00 | 1.00 | 1.00 |
| | 20000 | 8.94 | 106 | 108 | 25.10 | 25.48 | 2 | 1.53 | 2.07 | 2.99 | 2.22 | 2.24 |
| | 50000 | 23.39 | 644 | 648 | 67.49 | 68.07 | 5 | 4.01 | 12.57 | 17.85 | 5.96 | 5.98 |
| | 100000 | 40.92 | 2049 | 2043 | 161 | 159 | 10 | 7.01 | 40.01 | 56.26 | 14.19 | 13.97 |
| | 200000 | 89.31 | NA | NA | 387 | 399 | 20 | 15.29 | NA | NA | 34.21 | 35.04 |
| | 500000 | 225 | NA | NA | 1519 | 1492 | 50 | 38.61 | NA | NA | 134.22 | 131.09 |
| $d$ | 1000 | 16.13 | 3080 | 3131 | 1415 | 1510 | 1 | 1.00 | 1.00 | 1.00 | 1.00 | 1.00 |
| | 2000 | 23.22 | 5610 | 5665 | 1439 | 1537 | 2 | 1.44 | 1.82 | 1.81 | 1.02 | 1.02 |
| | 5000 | 45.99 | 13186 | 13240 | 1506 | 1678 | 5 | 2.85 | 4.28 | 4.23 | 1.06 | 1.11 |
| | 10000 | 84.25 | NA | NA | 1585 | 1698 | 10 | 5.22 | NA | NA | 1.12 | 1.12 |
| | 20000 | 169 | NA | NA | 1675 | 1714 | 20 | 10.53 | NA | NA | 1.18 | 1.14 |
| | 40000 | 320 | NA | NA | 2548 | 2043 | 40 | 19.84 | NA | NA | 1.80 | 1.35 |

*Table 15.* Ablation study.

| Datasets | CH | | DB | | NMI | | ACC | |
|---|---|---|---|---|---|---|---|---|
| | $\Psi$-DR | PP-DR | $\Psi$-DR | PP-DR | $\Psi$-DR | PP-DR | $\Psi$-DR | PP-DR |
| COIL20 | **3985** | 3178 | **0.72** | 1.20 | **0.960** | 0.927 | **0.997** | 0.979 |
| banknote | **37146** | 108 | **0.05** | 2.18 | **0.936** | 0.120 | **0.996** | 0.585 |
| landsat | **1990** | 1617 | **1.06** | 1.06 | **0.681** | 0.675 | **0.855** | 0.85 |
| pendigits | **7470** | 4651 | **1.27** | 2.00 | **0.819** | 0.791 | **0.906** | 0.905 |
| usps | **8253** | 6421 | **1.05** | 8.59 | 0.799 | **0.802** | **0.825** | 0.81 |
| letters | 418.739 | **509** | 11.917 | **11.45** | **0.513** | 0.484 | **0.607** | 0.576 |
| mnist | **52307** | 48181 | **0.96** | 1.64 | 0.814 | **0.815** | **0.97** | 0.841 |
| skin | **1279417** | 10467 | **0.13** | 3.47 | **0.828** | 0.011 | **0.996** | 0.984 |
| gisette_cos | **21224** | 655 | **0.34** | 3.06 | **0.655** | 0.063 | **0.931** | 0.596 |
| rcv1 | **15722** | 4210 | **15.73** | 21.23 | 0.343 | **0.348** | **0.547** | 0.499 |

