# OpenReview forum: "Dimensionality Reduction with Point-distributions Similarity Invariant"
_ICML.cc/2026/Conference — ICML 2026 regular_

### Official Review · Reviewer_qfMr · 2026-02-21

**Soundness:** 3
**Presentation:** 4
**Significance:** 3
**Originality:** 3
**Overall Recommendation:** 4
**Confidence:** 4

**Summary:**

The paper proposes a new dimensionality reduction technique designed to preserve point distribution.
It clusters the data with a linear-complexity technique, then maintains the distance distribution within and between clusters rather than all pairwise data, making the approach linear in the number of data. A kernel approach is used to encode the similarity between data and cluster distributions. Formal proofs of convergence are given, and experiments show the new approach outperforms baseline DR techniques.

Claims:
1) a new DR technique that preserve similarity between points and distribution
2) a linear time framework
3) formal proofs
4) extensive experimental validation

**Compliance With Llm Reviewing Policy:**

Affirmed.

**Final Justification:**

I acknowledge the authors' efforts to provide additional experimental results. I am not fully convinced by the lack of comparison with relevant approaches (landmark-based, faster by design). A discussion should clearly justify the absence of comparison to landmark-based approaches.

**Key Questions For Authors:**

1) What are the gaps in the recent literature in the domain of kernel approaches or other DR techniques that justify the proposal?
2) What are the hyperparameters of the proposed method, and how to tune them?
3) Is the approach shown in Figure 2 valid to tune the number of cluster k?
4) What happens if we use clustering techniques like HDBSCAN, which may provide non-linear clusters with a centroid outside the cluster support?

**Limitations:**

The impact statement is adequate for this paper.
But there is no discussion of the limitations:
1) The number of hyperparameters of the proposed method and the way to tune them.
2) Experiments only cover 2D projections; what about higher dimensions?
3) The approach is not better than PCA on single-cell RNAseq data, so what is the limit in terms of dimensionality and data size? What does that mean that PCA is better? Are clusters all well separated in these high-dimensional spaces? Refer to [H][I][J] papers.

**Strengths And Weaknesses:**

STRENGTHS
- The approach is novel in its use of a RKHS to preserve the distance distribution between points and clusters.
- The formal derivation of the approach is given
- The proof of convergence is given
- The paper is well written and clear.

WEAKNESSES
- Claim 4) The experimental part is not convincing as it does not involve strong competitors.
- Many details are missing to reproduce the experiment.
- Many relevant papers are missing, questioning the originality (Claims 1 and 2).


SOUNDNESS
- I do not have the technical skills to review Section 5 and the related appendix (I cannot judge claim 3).
Assuming this section 5 is valid, the paper proposes a novel approach to reduce the dimensionality with a strong formal basis. The main issue is then the experimental section, which does not compare to relevant competitors. The approach is a cluster-then-project approach [A]. The clustering part is independent of the projection part. Therefore:
1) The clustering part should be varied in the experiment to evaluate its importance. Other prototype-based approaches exist, of which many are linear in the number of data points: for instance, K-means and its variants or Gaussian Mixture Models.

2) To evaluate the projection part of PSIDR alone, without the clustering, we can use the data labels directly as cluster labels. The projection evaluation with clustering metrics would still be valid, as it compares the clustering results running K-means in the projection space to the data labels.

3) The projection part of PSIDR is a supervised projection having access to both pairwise distances and class information (Algorithm 1, instruction 3) and is evaluated based on this clustering information, while competitors PCA, tSNE, UMAP and AE do not have access to the clustering result. Valid competitors should be supervised DR techniques, for instance [B] and [C].

[A] Towards a Systematic Combination of Dimension Reduction and Clustering in Visual Analytics
IEEE TVCG 2018 DOI: 10.1109/TVCG.2017.2745258

[B] Supervised version of UMAP: https://umap-learn.readthedocs.io/en/latest/supervised.html

[C] Supervised tSNE: Steering Distortions to Preserve Classes and Neighbors in Supervised Dimensionality Reduction
NeurIPS 2020
https://zenodo.org/records/4094851

- The linear time complexity is one claim of the work. The competitors are the base versions, while more recent versions either better preserve cluster structure (PacMAP) [D] or are faster (Barnes-Hut t-SNE) [E]. Moreover, landmark-based approaches are closely related to the idea of measuring distance to k cluster centers; for instance, Landmark-ISOMAP [E'] has an N log(N) complexity.

[D] Understanding How Dimension Reduction Tools Work: An
Empirical Approach to Deciphering t-SNE, UMAP, TriMap,
and PaCMAP for Data Visualization
JMLR 2021
https://jmlr.org/papers/volume22/20-1061/20-1061.pdf
https://github.com/YingfanWang/PaCMAP?tab=readme-ov-file

[E] Accelerating t-SNE using Tree-Based Algorithms.
JMLR 2014
https://jmlr.org/papers/v15/vandermaaten14a.html

[E'] Global Versus Local Methods in Nonlinear Dimensionality Reduction
NeurIPS 2002
https://proceedings.neurips.cc/paper/2002/hash/5d6646aad9bcc0be55b2c82f69750387-Abstract.html

- The evaluation relies on labeled benchmark datasets to provide a ground truth to evaluate the quality of the projection in terms of clustering and classification. When using labeled data for clustering evaluation, it is important to first measure the clustering quality of the ground truth. Classes in certain datasets do not form well-separated clusters [F, G]. Exploring Table 9, I could find that the Banknote dataset is ranked 42/96 with score 0.197 (1 best, 0 worst) in paper [G], so certain of its classes do not form well-clustered groups. Landsat, USPS, Skin, Gisette, Rcv1 are not evaluated in [G], but the code therein could be used to check their quality.

[F] Classes are Not Clusters: Improving Label-Based Evaluation of Dimensionality Reduction
IEEE TVCG 2024
https://doi.org/10.1109/tvcg.2023.3327187

[G] Measuring the Validity of Clustering Validation Datasets
IEEE TPAMI 2025
10.1109/TPAMI.2025.3548011
https://github.com/hj-n/labeled-datasets

- The linear complexity is not validated with experimental data. We should have a line chart showing computing time as a function of the number of data points.

- It seems PCA is the best for single-cell expression data for both time and clustering preservation. What is the limit of the proposed approach in terms of dimensionality?

- Hyperparameters are missing:

1) PSIDR requires a kernel to be characteristic. What are the kernel \kappa and function \phi used for the PSIDR technique?
2) How do we set the number of clusters k? Figure 2 seems to indicate that we can run a grid search and keep the k that gives the best prediction clustering score. But no clear guideline is given. Has this analysis been done for other datasets? Is it a valid approach to tune k?
3) The hyperparameters of tSNE and UMAP are not given. It is well known that these parameters and the initialization strongly impact the clustering of the projection [H,I,J]. What are the hyperparameters, and how were they selected?

[H] The art of using t-SNE for single-cell transcriptomics
https://www.nature.com/articles/s41467-019-13056-x
https://github.com/berenslab/rna-seq-tsne

[I] From t-SNE to UMAP with contrastive learning ICLR 2023
https://openreview.net/forum?id=B8a1FcY0vi

[J] The art of seeing the elephant in the room: 2D embeddings of single-cell data do make sense
https://doi.org/10.1371/journal.pcbi.1012403

- I would like to see the 2D plots for all datasets and techniques in the appendix to better understand what the summary scores mean in terms of cluster patterns.

PRESENTATION

Overall, the paper is very clear and easy to follow, aside from the missing hyperparameters specification.

SIGNIFICANCE

The problem of dimensionality reduction is quite mature; this contribution is interesting but not particularly significant unless it proves competitive in terms of computing time and accuracy against state-of-the-art competitors, on a fair basis (supervised techniques, accessing the same cluster information), or it provesa  clear improvement over other DR techniques using the kernel trick (see below).

ORIGINALITY

The paper explores a new approach to dimensionality reduction using RKHS.
But it lack discussion of prior work in this domain:

[K] Dimensionality Reduction for Supervised Learning with Reproducing Kernel Hilbert Spaces
JMLR 2004
https://www.jmlr.org/papers/volume5/fukumizu04a/fukumizu04a.pdf

[L] Solving Interpretable Kernel Dimensionality Reduction
NeurIPS 2019
https://papers.nips.cc/paper_files/paper/2019/hash/61f3a6dbc9120ea78ef75544826c814e-Abstract.html

and it misses recent surveys on dimensionality reduction techniques:

[M] Toward a Quantitative Survey of Dimension Reduction Techniques IEEE TVCG 2021
10.1109/TVCG.2019.2944182

[N] Multidimensional Projection for Visual Analytics: Linking Techniques with Distortions, Tasks, and Layout Enrichment IEEE TVCG 2018
10.1109/TVCG.2018.2846735

I recognise the large effort put into the formal demonstrations. The main issues in the paper are the limited experimental validation using low-hand competitors. The paper could focus only on the formal aspect (Claim 3), toning down Claim 4, but then we would lack a deep analysis of the related work on the RKHS approach to show where the gaps are and justify the approach.

MINOR

- The name of the proposed technique in Figure 1 is wrong. Also mention "Ours" near its name to identify yours.
- L211 mentions "four stages," but R112 mentions a "two-stage" procedure, and I had in mind the pre-processing clustering procedure, so "3-stage". This is confusing.
- R149: We use the term "stress" rather than "strain" for the loss minimized during a projection.
- missing technical references for:
1) the "approximate square distance" (L167)
2) the "closed-form solution" (R153) of equation (1).
3) classical MDS (R142)

- The algorithm 1 request \kappa as input, but \kappa, and instructions 3 and 9 depend on \phi. Hence, we also need to specify \phi as input.

- The three footnotes on page 4 should be removed. The link to the source code can be an additional reference and be mentioned right at the end of the introduction. The details about distance d can be included in the text only when necessary, either near L167 or near Lemma 5.6.

- Table 4 should indicate the dimensionality and number of data points in each dataset to help analyze the results.

---

> ### Author Rebuttal · Authors · 2026-03-30
>
> Thank you for your review and constructive feedback.
>
> We focused our comparisons on unsupervised dimensionality reduction methods, whereas many of the techniques you suggested, such as [B, C, K], are supervised. Consequently, we did not include them in our comparative analysis. Instead, we compared PaCMAP (the latest algorithm in the PaCMAP suite) against Neg-t-SNE, InfoNC-t-SNE, and LocalMAP (https://github.com/williamsyy/LocalMAP).  (Due to character limitations, please refer to the results presented in the response to reviewer #wsHb.)
>
> The experimental results demonstrate that our method outperforms state-of-the-art approaches.
>
> We compared their running times on the rcv1 dataset across various random sampling points and dimensions, as follows:
>
> |  |  |  | Time  | |  |    |    |   | Ratio   |    |    |
> |-|-|-|-|-|-|-|-|--|-|-|-|
> | n | $\Psi$-DR | Neg-t-SNE | InfoNC-t-SNE | PaCMAP  | LocalMAP | n | $\Psi$-DR | Neg-t-SNE | InfoNC-t-SNE | PaCMAP| LocalMAP |
> | 10000| 5.84| 51.23| 36.32| 11.32 | 11.38 | 1| 1| 1| 1| 1| 1 |
> | 20000 | 8.94| 105.83| 108.43| 25.10 | 25.48| 2 | 1.53| 2.07 | 2.99 | 2.22  | 2.24 |
> | 50000| 23.39| 644.14| 648.37| 67.49 | 68.07 | 5 | 4.01| 12.57  | 17.85 | 5.96 | 5.98 |
> | 100000| 40.92| 2049.46| 2043.45  | 160.66| 158.98 | 10 | 7.01   | 40.01 | 56.26 | 14.19  | 13.97 |
> | 200000 | 89.31|  | | 387.27| 398.74 | 20 | 15.29 | | | 34.21 | 35.04|
> | 500000| 225.49 | || 1519.42 | 1491.77  | 50 | 38.61 |  | | 134.22  | 131.08  |
>
> |  |  | | Time |  |  | | | | Ratio | | |
> |-|-|-|-|-|-|-|-|-|-|--|--|
> | d  | $\Psi$-DR| Neg-t-SNE | InfoNC-t-SNE | PaCMAP  | LocalMAP | d  | $\Psi$-DR      | Neg-t-SNE | InfoNC-t-SNE | PaCMAP   | LocalMAP |
> | 1000 | 16.13  | 3080.42   | 3130.95      | 1415.27 | 1509.87  | 1  | 1            | 1               | 1            | 1                | 1        |
> | 2000  | 23.22  | 5610.18   | 5665.32      | 1438.89 | 1536.90   | 2  | 1.44        | 1.82        | 1.81         | 1.02          | 1.02 |
> | 5000  | 45.99  | 13185.76  | 13240.08     | 1506.04 | 1678.30   | 5  | 2.85      | 4.28        | 4.23     | 1.06              | 1.11 |
> | 10000 | 84.25  | |  | 1585.29 | 1698.07   | 10 | 5.22    |         |             | 1.12              | 1.12 |
> | 20000 | 169.77 | | | 1674.69 | 1714.23  | 20 | 10.53    |                  |               | 1.18        | 1.14    |
> | 40000 | 320.06 | | | 2548.12 | 2042.99  | 40 | 19.84     |                  |             | 1.80          | 1.35   |
>
> Our method is linear with respect to both points $n$ and dimensions $d$, while Neg-t-SNE and InfoNC-t-SNE are linear with respect to dimensions $d$, but have greater than linear time complexity with respect to the number of points $n$. PaCMAP and LocalMAP also have greater than linear time complexity with respect to the number of points $n$, but they are less affected by the number of dimensions $d$.
>
> We will plot graphs illustrating the running times and ratios for varying numbers of points ($n$) and dimensions ($d$) to provide a more intuitive visualization, and report the exact runtime values ​​in the appendix.
>
> Existing algorithms overlook the information that the data is generated from $k$ distinct distributions; by leveraging this information, we introduce the concept of "point-distribution similarity invariant." Furthermore, we propose an algorithm that preserves this invariant to achieve dimensionality reduction. The proposed method constitutes an efficient dimensionality reduction algorithm with linear complexity in terms of both $n$ and $d$.
>
> We provide details regarding the kernel functions and parameters used in Appendix J: "Experimental Settings." Our experiments demonstrate that effective dimensionality reduction is achieved when the number of distributions aligns with the number of clusters, $k$; consequently, we can utilize the spectral gap method, as described in the KBC paper, to determine the value of $k$. Alternatively, we may employ the method outlined in the paper cited by Reviewer #bgi1 to determine $k$.
>
> For complex datasets, KBC and HDBSCAN may produce cluster centers that lie outside the clusters; however, this poses no issue, as our approach relies on the data distribution (via kernel mean embedding) rather than on the specific location of the center points.
>
>
> We are deeply grateful for your suggestions, including those not explicitly mentioned above. In addition to the papers concerning the comparative algorithms discussed above, we will also cite and discuss the other papers you mentioned. Due to character limitations, we are unable to provide a point-by-point response here; however, we will address all of these matters in the final version. For instance, we have implemented changes such as removing the footnote on page 4, placing the link to the source code at the end of the Introduction, and indicating the dimensionality and number of data points in each dataset in Table 4. We have made these revisions and will present them in the revised version.

---

> > ### Author Rebuttal · Reviewer_qfMr · 2026-04-02
> >
> > I acknowledge the authors' responses to my comments and the additional experiments proposed.
> > I will raise my score accordingly but I think the paper still needs more work on the experimental side (ablation study, landmark approaches...) and the discussion of relevant related work that I suggested.

---

> > > ### Author Response · Authors · 2026-04-04
> > >
> > > We are pleased that you were satisfied with our previous experiments, and we thank you for your suggestions regarding the ablation experiments.
> > >
> > > Our contribution is proposing a point-distribution similarity invariant, so we conducted the following ablation experiments. We compared the results of maintaining this invariant ( $\Psi$-DR) with maintaining the point-point similarity invariant (PP-DR). The point-point similarity invariant reduces dimensionality by preserving the distance between each point in $C_c$ (the second line of the algorithm) and the center of $C_c$. The results are shown below:
> > >
> > > | Datasets    |    CH    |          |   DB   |        |   NMI  |       |   ACC  |       |
> > > |-------------|:--------:|:--------:|:------:|:------:|:------:|:-----:|:------:|:-----:|
> > > |             |  $\Psi$-DR  |   PP-DR  |  $\Psi$-DR |  PP-DR |  $\Psi$-DR | PP-DR |  $\Psi$-DR | PP-DR |
> > > | COIL20      | 3985 | 3178 | 0.72   | 1.20  | 0.960   | 0.927 | 0.997  | 0.979 |
> > > | banknote    | 37146 | 107 | 0.05  | 2.18  | 0.936  | 0.120  | 0.996  | 0.585 |
> > > | landsat     | 1990 | 1617 | 1.06  | 1.06  | 0.681  | 0.675 | 0.855  | 0.850  |
> > > | pendigits   | 7470 | 4651 | 1.27  | 2.00  | 0.819  | 0.791 | 0.906  | 0.905 |
> > > | usps        | 8252 | 6421 | 1.05  | 8.59  | 0.799  | 0.802 | 0.825  | 0.810  |
> > > | letters     | 418  | 509  | 11.92 | 11.45 | 0.513  | 0.484 | 0.607  | 0.576 |
> > > | mnist       | 52306 | 48181 | 0.96  | 1.64  | 0.814  | 0.815 | 0.970   | 0.841 |
> > > | skin        | 1279417  | 10467    | 0.13  | 3.47  | 0.828  | 0.011 | 0.996  | 0.984 |
> > > | gisette | 21224 | 655.025  | 0.34   | 3.06  | 0.655  | 0.063 | 0.931  | 0.596 |
> > > | rcv1        | 15722 | 4210.138 | 15.73 | 21.23 | 0.343  | 0.348 | 0.547  | 0.499 |
> > >
> > >
> > > The results demonstrate that our proposed point-distribution similarity invariant is effective.
> > >
> > > We will add this experiment, along with the experiments shown in rebuttal, to the paper. We will also cite and discuss the papers you mentioned, as we responded in our rebuttal. We hope these new results satisfactorily address your concern.

---

### Official Review · Reviewer_bgi1 · 2026-03-10

**Soundness:** 2
**Presentation:** 4
**Significance:** 3
**Originality:** 3
**Overall Recommendation:** 4
**Confidence:** 3

**Summary:**

The paper proposes a new method of "similarity preservation" for dimension reduction methods, where the similarity structure between data points and their distributions are preserved by using RKHS techniques in Algorithm 1 ($\Psi$-DR), which runs in time linear to data input. Under some assumptions, the $\Psi$-DR is asymptotically consistent (Theorem 5.7), and the authors also give a convergence rate in Theorem 5.8. Experiments are done to compare $\Psi$-DR with several metrics, with $\Psi$-DR outperforming several dimension reduction algorithms on a a wide variety of datasets.

**Compliance With Llm Reviewing Policy:**

Affirmed.

**Final Justification:**

My score remains unchanged.

My main concern is still not answered, and I don't know if it's because I asked the follow-up question wrongly, or otherwise.

I still don't understand why the equality holds

$\frac{1}{n} \sum_{i=1}^{n} |z_i - z_i^{\ast}|_2^2$

$=\frac{1}{n} \sum_{i=1}^n |z_i - z_i^{\ast}{\mathbf Q}|_2^2$

That is to say, why can we replace $z_i^{\ast}$ with $z_i^{\ast}Q$? As in, I understand $z_i$ and $z_i^{\ast}$ are not aligned, and we rotate $z_i^{\ast}$ with $Q$. Why does this give equivalence?

I guess in my mind, I am picturing two vectors in 2D, e.g. hands of a clock, and find the distance between them. Why would the hands have the same distance if one hand has been rotated?

If this inequality does not hold, then the rate of convergence theorem is false.

This does not change the main result, but it would be significantly weaker without a correct convergence rate theorem. On the other hand, it is also possible I may have misread the proof.

Therefore, I maintain my score as both outcomes could be equally likely.

**Key Questions For Authors:**

1. My main question is the proof of Theorem 5.8 (see above), where I am probably missing something with that equation. Could the authors clarify?

2. The last paper I reviewed a year ago

https://arxiv.org/pdf/2311.16614

assesses unimodality, but could be wrapped into an algorithm to detect the number of clusters in a dataset. I'm not asking for any experiments (although I think the authors of that paper have released some code), but I'm wondering how the algorithm in that paper would compare with ($\Psi$-DR)? What would be the dimensionality reduction invariant that is preserved in that paper?


Answering Q1 would possibly change my score upward.

**Limitations:**

Yes, but in Appendix I. Since this is just two lines, I would suggest moving the limitations to the main paper after the Experiments section.

**Strengths And Weaknesses:**

In general, I find this to be a thorough, well-written paper where I learnt something new (while I work with dimension reduction algorithms, I do not think about categorizing what is preserved when these algorithms are used, which the authors term as the dimension reduction invariant). The paper is clear with the authors giving intuition behind their theorems, as well as explaining each step of their results, with extensive experiments validating their results. I have checked through the proofs as best as I could (with a concern below on the convergence rate [Theorem 5.8 proof]), and believe they are mostly correct.

**Soundness:**

The claims appear to be well-supported, in the sense that it is easy to follow the author's reasoning in most proofs, which is much appreciated. The assumptions seem reasonable. I have tried my best to validate the proofs given in the last week or so. However, I do not understand the proof of Theorem 5.8 on Page 17, as the equation on lines 926-928 states

$\frac{1}{n} \sum_{i=1}^{n} \|z_i - z_i^{\ast}\|_2^2$

$=\frac{1}{n} \sum_{i=1}^n \|z_i - z_i^{\ast}{\mathbf Q}\|_2^2$.

Why does this hold? [also, are you averaging over $i$ or $n$?) I can understand if the second term is $\frac{1}{n}\sum_{i=1}^n \|z_i{\mathbf Q} - z_i^{\ast}{\mathbf Q}\|_2^2$ since ${\mathbf Q}$ is orthonormal, but I don't understand why the equality holds as it is written. What am I missing here?

Other than that, the experiments appear to be well-designed.

**Presentation:** The presentation of this paper is extremely clear and well-structured. What stands out to me is the introduction which gives sufficient motivation for the direction that the authors take, as well as the explanation given behind each lemma and theorem. The appendix is sufficiently detailed as well.

**Significance**: The construction of $\Psi$-DR is significant to me, and could lead to other researchers building on this work to create dimension reduction algorithms that preserves this particular invariant.

**Originality**: I like how the authors coin the term "dimensionality reduction invariant", which characterizes what different dimension reduction algorithms preserve. I find this new and a different way of thinking of these algorithms. The work designs the first algorithm to preserve the similarity structure between data points and distributions they come from.

---

> ### Author Rebuttal · Authors · 2026-03-30
>
> Thank you for your review and insightful comments.
>
> A1. Thank you for the careful reading and pointing out the ambiguity on Line 926. Our analysis there specifically addresses the error after alignment has been performed (by Q); consequently, the expression $\frac{1}{n}\sum_{i=1}^n \|z_i - z_i^*\|^2$  becomes $\frac{1}{n}\sum_{i=1}^n \|z_i - z_i ^*Q \|^2$ following alignment by $Q$.
>
> In the revised version, we will add further clarifications regarding this theorem and its proof, specifically concerning the form of $z^*$ after $Q$-alignment, to prevent any future misunderstandings.
>
> A2. MUD-POD attempts to identify the projection direction that best captures the structural characteristics of a distribution by randomly projecting data onto a one-dimensional space; this direction then serves as a benchmark for detecting multimodality. When our method is applied, the invariant between individual data points and their underlying distributions is perfectly preserved. Consequently, within the projected space, points belonging to the same distribution should remain tightly clustered around their respective distribution centers, which is a condition that satisfies the unimodality assumption central to the MUD-POD test. Specifically, MUD-POD is designed to determine whether a dataset deviates from the hypothesis of a single underlying distribution (i.e., to detect the presence of multiple distinct distributions), whereas our method serves the purpose of dimensionality reduction while ensuring the preservation of this point-to-distribution invariant after dimensionality reduction. Our method requires a priori specification of the number of underlying distributions, which is a requirement that MUD-POD is uniquely capable of fulfilling.
>
> In the revised version of the manuscript, we will relocate the "Limitations" section to follow the "Experiments" section within the main text.

---

> > ### Author Rebuttal · Reviewer_bgi1 · 2026-03-31
> >
> > Could you elaborate on why this leads to an equality please? I will also take a second look at the paper after the elaborations to understand this.
> >
> > That being said, the proof of convergence rate does not affect the main claim of this paper.

---

> > > ### Author Response · Authors · 2026-04-02
> > >
> > > Thank you for your reply.
> > >
> > > We wanted to analyze the error in the dimensionality reduction result. However, due to the rotation invariance of the dimensionality reduction result, the $z_i$ and $z_i^\*$ results are not aligned. To remove the error caused by rotation invariance, we consider the result after $z_i$ and $z_i^\*$ are aligned, that is, the result of $z_i^*$ after $Q$ rotation transformation, i.e., $\frac{1}{n} \sum_{i=1}^n\|z_i−z_i^∗\|^2  \rightarrow \frac{1}{n} \sum_{i=1}^n\|z_i−z_i^∗Q\|^2$.

---

### Official Review · Reviewer_Fmcz · 2026-03-11

**Soundness:** 3
**Presentation:** 3
**Significance:** 3
**Originality:** 3
**Overall Recommendation:** 4
**Confidence:** 5

**Summary:**

The authors proposed a dimension reduction (DR) method preserving the invariant of the point-distribution similarity. This method is computationally efficient. Some theoretical properties are investigated. Numerical experiments are implemented to demonstrate the performance of the proposed method.

**Compliance With Llm Reviewing Policy:**

Affirmed.

**Key Questions For Authors:**

-In the Figure 1, is the name ‘D-split’ of subfigure (f) incorrect?

-Some notations seems to be not used. For example, $\rho$ in Line 93 of Page 2.

**Limitations:**

yes

**Strengths And Weaknesses:**

Strengths:

The authors proposed a new dimension reduction method suitable for cluster structure, and designed a new invariant principle, the point-distributions similarity. The claim seems reasonable.Evaluations seems sound. The proposed PSIDR outperforms other methods mainly on the metrics (Calinski-Harabasz, Davies-Bouldin, and Normalized Mutual Information). I am concern that the $k$ subsets required by the algorithm are provided by the KBC method (Zhang et al. 2025a), but the theoretical results seems unrelated to the initial clustering result.


Weakness:
-As pointed out by the authors, the proposed method is not suitable for unbalanced cases.

-As I mentioned before, the initial data partition depends on some clustering algorithms (e.g., KBC used here). Moreover, the quality of the clustering results determines the effectiveness of the dimensionality reduction. This is unwanted in my opinion.

---

> ### Author Rebuttal · Authors · 2026-03-30
>
> Thank you for your review and valuable suggestions.
>
>
> A1. Thank you for pointing out the typo in Figure 1. We have corrected "D-split" to "$\Psi$-DR".
>
> A2. We now change "Formally, let $\rho: \mathcal{P}(\mathcal{X}) \times \mathcal{P}(\mathcal{X}) \to [0,1]$ be a similarity function satisfying $\rho(\nu,\nu') = \langle \nu, \nu' \rangle_{\mathcal{H}}$ for the Hilbert space $\mathcal{H}$" to "Formally, we quantify the similarity between $\nu, \nu' \in \mathcal{P}(\mathcal{X})$ using the inner product $\langle \nu, \nu' \rangle_{\mathcal{H}}$ in a Hilbert space $\mathcal{H}$".

---

> > ### Author Rebuttal · Reviewer_Fmcz · 2026-04-03
> >
> > I am satisfied with the revision.

---

### Official Review · Reviewer_wsHb · 2026-03-17

**Soundness:** 3
**Presentation:** 3
**Significance:** 3
**Originality:** 2
**Overall Recommendation:** 3
**Confidence:** 3

**Summary:**

The paper proposes a new dimensionality reduction model, and a linear-time method to achieve this reduction. The paper also derives theoretical guarantees for their dimensionality reduction technique, and presents experiments on several datasets.

**Compliance With Llm Reviewing Policy:**

Affirmed.

**Key Questions For Authors:**

See weaknesses

**Limitations:**

See weaknesses

**Strengths And Weaknesses:**

Strengths:

- New idea, to the best of my knowledge.
- Important theoretical guarantees.
- The paper is well presented.

Weaknesses:
- The experimentation seems somewhat outdated, both in terms of datasets and in terms of comparative baselines. The most recent method they compare against is UMAP, which is already 8 years old. Therefore, the performance of this method is somewhat unclear: how will it compare against more modern methods, on more sophisticated datasets?

Minor comment (but important typo): Figure 1 does not include PSIDR. I'm guessing it is D-Split? This is a somewhat important typo.

---

> ### Author Rebuttal · Authors · 2026-03-30
>
> Thank you for your review and suggestions.
>
> We have now compared the following four methods: PaCMAP[1], Neg-t-SNE[2], InfoNC-t-SNE[2], and LocalMAP[3].
>
> [1] Wang Y, Huang H, Rudin C, et al. Understanding how dimension reduction tools work: an empirical approach to deciphering t-SNE, UMAP, TriMAP, and PaCMAP for data visualization[J]. Journal of Machine Learning Research, 2021, 22(201): 1-73.
>
> [2] Damrich, S.; Böhm, J. N.; Hamprecht, F. A.; and Kobak, D. From t-SNE to UMAP with contrastive learning. In International Conference on Learning Representations. 2023.
>
> [3] Wang Y, Sun Y, Huang H, et al. Dimension reduction with locally adjusted graphs[C]//Proceedings of the AAAI Conference on Artificial Intelligence. 2025, 39(20): 21357-21365.
>
> ||||CH|||||DB|||
> |-|-|-|-|-|-|-|-|-|-|-|
> |Datasets|$\Psi$-DR|Neg-t-SNE|InfoNC-t-SNE|PaCMAP|LocalMAP|$\Psi$-DR|Neg-t-SNE|InfoNC-t-SNE| PaCMAP|LocalMAP|
> |COIL20|3985|626| 953|7758|2787|0.72|3.87|4.19|3.69|4.58|
> |banknote|37146|552|752|747|714|0.05|1.40|1.21|1.10|1.21|
> |landsat|1990|1669|1183|1892|1108|1.06|1.03|1.10|1.05|1.09|
> |pendigits|7470|4037|3785|4357|3379|1.27| 3.72| 2.24 | 2.68|4.32|
> |usps|8252|5012|3957|8192|6374|1.05| 1.93 | 1.96 | 1.43  | 1.11   |
> |letters|418|715| 502|541|576|11.92|11.01  | 17.19  | 16.18 | 9.81|
> |mnist|52306|46188|25404|81764|46269|0.96| 1.48  | 1.76 | 1.29  | 0.87 |
> |skin|1279417|6475|3886|47859|44706|0.13| 3.62 | 4.82  | 1.46  | 1.48|
> |gisette|21224|6171|5549|19091|9896|0.34| 0.98  | 1.02  | 0.51  | 0.70 |
> |rcv1|15722|NA|NA|17381|13111|15.73|NA|NA| 21.28 | 13.17  |
>
> |  |       |     | NMI    |   |    |    |     | ACC  |        |     |
> |---|--|---|--|---|---|---|----|-----|--|---|
> | Datasets  | $\Psi$-DR   | Neg-t-SNE | InfoNC-t-SNE | PaCMAP | LocalMAP | $\Psi$-DR   | Neg-t-SNE | InfoNC-t-SNE | PaCMAP | LocalMAP |
> | COIL20| 0.960  | 0.826   | 0.831        | 0.910   | 0.900      | 0.997 | 0.878     | 0.868        | 0.875  | 0.885    |
> | banknote | 0.936 | 0.603  | 0.701        | 0.392  | 0.447    | 0.996 | 0.996     | 0.993        | 1.000      | 1.000    |
> | landsat   | 0.681 | 0.633  | 0.576        | 0.690   | 0.643    | 0.855 | 0.848     | 0.853        | 0.855  | 0.870     |
> | pendigits | 0.819 | 0.818   | 0.733    | 0.872  | 0.832    | 0.906 | 0.982     | 0.974        | 0.987  | 0.991    |
> | usps    | 0.799 | 0.663  | 0.602      | 0.756  | 0.815    | 0.825 | 0.874     | 0.870         | 0.918  | 0.952    |
> | letters   | 0.513 | 0.499  | 0.435     | 0.520   | 0.504    | 0.607 | 0.797     | 0.789        | 0.790   | 0.852    |
> | mnist   | 0.814 | 0.630   | 0.600    | 0.740   | 0.807    | 0.970  | 0.876     | 0.853        | 0.919  | 0.933    |
> | skin  | 0.828 | 0.086  | 0.047    | 0.022  | 0.035    | 0.996 | 0.998     | 0.989        | 0.998  | 0.999    |
> | gisette | 0.655 | 0.678   | 0.687   | 0.736  | 0.690     | 0.931 | 0.952     | 0.959        | 0.975  | 0.981    |
> | rcv1  | 0.343 | NA    | NA     | 0.360   | 0.336    | 0.547 | NA        | NA           | 0.620   | 0.656    |
>
> |    | $\Psi$-DR | Neg-t-SNE | InfoNC-t-SNE | PaCMAP | LocalMAP |
> |:-:|:-:|:-:|:-:|:-:|:-:|
> | winner  |18 | 2  | 0  | 9 | 12   |
> | runner-up |11 | 5  | 2  | 15| 6 |
>
> | Runtime     | Datasets    | $\Psi$-DR     | Neg-t-SNE | InfoNC-t-SNE | PaCMAP  | LocalMAP |
> |-------------|-------------|---------|-----------|--------------|---------|----------|
> | COIL20      | COIL20      | 0.66   | 3.72      | 1.35         | 21.62   | 14.09    |
> | banknote    | banknote    | 0.15   | 1.25      | 1.28         | 13.45   | 16.05    |
> | landsat     | landsat     | 0.16   | 1.16      | 2.30          | 14.54   | 13.80     |
> | pendigits   | pendigits   | 2.19    | 7.39      | 7.83         | 20.28   | 22.30     |
> | usps        | usps        | 6.32   | 8.86      | 9.15         | 24.5    | 26.05    |
> | letters     | letters     | 10.47   | 13.07     | 14.27        | 32.95   | 33.40     |
> | mnist       | mnist       | 28.55  | 66.29     | 70.52        | 87.28   | 82.90     |
> | skin        | skin        | 22.71  | 175    | 187       | 332  | 339.43   |
> | gisette | gisette_cos | 2.92    | 9.08      | 9.66         | 26.39   | 25.51    |
> | rcv1        | rcv1        | 382 | NA        | NA           | 2911 | 2809  |
>
>
>
> The experimental results demonstrate that our method outperforms state-of-the-art approaches.
>
> We compared their running times on the rcv1 dataset across various random sampling points and dimensions (Due to character limitations, please refer to the results presented in the response to reviewer #qfMr.)
>
> Our method is linear with respect to both points $n$ and dimensions $d$, while Neg-t-SNE and InfoNC-t-SNE are linear with respect to dimensions $d$ but have greater than linear time complexity with respect to the number of points $n$. PaCMAP and LocalMAP also have greater than linear time complexity with respect to the number of points $n$, but they are less affected by the number of dimensions $d$.
>
>
> Thank you for pointing out the typo in our Figure 1. We have now corrected it.

---

### Decision · Program_Chairs · 2026-04-30

**Decision:**

Accept (regular)

**Comment:**

This paper proposes a novel dimensionality reduction framework that preserves point–distribution similarity. The problem is well motivated, and several reviewers highlighted both the clarity of the presentation and the potential significance of the work. The rebutta incorporates more recent baselines and addressing several presentation concerns. While some issues remain, including partial ambiguity in  implementation details, these appear addressable in the camera-ready version. Overall, considering the novelty of the perspective and the generally positive reviewer feedback, I am inclined to recommend acceptance.